# Bounds on quantum evolution complexity via lattice cryptography

**Ben Craps[1][⋆], Marine De Clerck[1][†], Oleg Evnin[2,1][‡], Philip Hacker[1][∘] and Maxim Pavlov[1][§]**

**1** Theoretische Natuurkunde, Vrije Universiteit Brussel (VUB) and
The International Solvay Institutes, Pleinlaan 2, B-1050 Brussels, Belgium
**2** Department of Physics, Faculty of Science, Chulalongkorn University,
Thanon Phayathai, Bangkok 10330, Thailand

⋆ Ben.Craps@vub.be , † Marine.Alexandra.De.Clerck@vub.be , ‡ oleg.evnin@gmail.com ,
∘ Philip.Hacker@vub.be , § Maxim.Dmitrievich.Pavlov@vub.be

## Abstract

We address the difference between integrable and chaotic motion in quantum theory as manifested by the complexity of the corresponding evolution operators. Complexity is understood here as the shortest geodesic distance between the time-dependent evolution operator and the origin within the group of unitaries. (An appropriate 'complexity metric' must be used that takes into account the relative difficulty of performing 'nonlocal' operations that act on many degrees of freedom at once.) While simply formulated and geometrically attractive, this notion of complexity is numerically intractable save for toy models with Hilbert spaces of very low dimensions. To bypass this difficulty, we trade the exact definition in terms of geodesics for an upper bound on complexity, obtained by minimizing the distance over an explicitly prescribed infinite set of curves, rather than over all possible curves. Identifying this upper bound turns out equivalent to the closest vector problem (CVP) previously studied in integer optimization theory, in particular, in relation to lattice-based cryptography. Effective approximate algorithms are hence provided by the existing mathematical considerations, and they can be utilized in our analysis of the upper bounds on quantum evolution complexity. The resulting algorithmically implemented complexity bound systematically assigns lower values to integrable than to chaotic systems, as we demonstrate by explicit numerical work for Hilbert spaces of dimensions up to $\sim 10^4$.

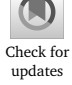
# 1 Introduction

If every computation that runs is a physical process, it is tempting to regard every physical process as a running computation. Then, so viewed, is it a difficult computation, or an easy one? The notion of *complexity* has been considered for classical dynamical trajectories, or even dynamical sequences [1], with a number of intricacies and ambiguities involved. Here, we shall be concerned with concrete proposals to estimate the complexity of quantum evolution.

If we are to view a dynamical quantum process as a computation, it should logically be a quantum computation. Complexity of quantum computations is a cornerstone topic in quantum information theory [2]. In application to the evolution of continuous physical systems, as opposed to discrete quantum computers, it has often been brought up in recent literature on high-energy theory subjects, in particular, in relation to black hole physics, spurred by the considerations of [3–6]. For computations that run on conventional quantum computers, complexity is defined in terms of the number of simple elementary operations ('gates') one must apply sequentially to reach the desired result. Generalizing this notion to continuous physical systems, and especially quantum field theories, meets considerable difficulties [8–10].

Apart from the proliferation of degrees of freedom and infinite-dimensional Hilbert spaces, multiple choices are involved in adapting the notion of complexity to the evolution of general physical systems. One such choice is whether the notion of being complex or simple should be attached to physical states, or to the unitary evolution operators. If it is the complexity of unitary evolution operators that is getting assessed, one must decide with what precision one is willing to approximate them based on operators from some pre-defined set, and from which precise set. A short contemporary review discussing these choices can be consulted in [11].

One notion of complexity that we find particularly attractive for applying to quantum dynamics is Nielsen's complexity [12–14]. This quantity is defined for any unitary operator, and hence for evolution operators, and is computed as the length of the shortest geodesic on the group of unitaries that connects the identity to the unitary operator in question. For defining the geodesics, one typically does not rely on the usual bi-invariant metric on the group of unitaries, but rather introduces 'cost factors' that scale up the contributions to the line element coming from a subset of 'hard' group generators, as opposed to the remaining 'easy' group generators. The notions of hard and easy come physically from the greater difficulty associated with implementing nonlocal operations than local ones. In practice, one chooses the easy (local) generators to be the operators that act only on a small number of particles/spins (depending on the concrete system under consideration), and hard (nonlocal) generators to be those that act simultaneously on a large number of particles/spins. A variety of prescriptions for assigning greater cost to nonlocal operations can in principle be devised, see for instance [15], sometimes referred to as the 'penalty schedule' [16].

How do evolution operators of physical dynamical systems behave with respect to this notion of complexity? It is logical to think of the Hamiltonian $H$ as a local (easy) operator, which makes the evolution trajectory $e^{-iHt}$ a geodesic on the group of unitaries. Thus at early times, complexity as defined above will be given simply by the length of this geodesic, and will grow linearly with time. At later moments, this geodesic will however cease to be the global distance minimizer and will have to be replaced by another shorter geodesic for the purpose of complexity estimation. It is furthermore expected that the complexity growth will saturate at a plateau [15,17], followed by sharp decreases due to Poincaré revivals (with $e^{-iHt}$ passing close to the identity) at much later, essentially unobservable times.

Does this notion of complexity distinguish between generic systems and those systems that are 'solvable' and display analytic structures such as integrability? Intuitively, solvability means precisely that the entire possible range of motions can be expressed through a small set of functions. This is exactly what happens when an analytic solution for a dynamical problem is written down. For instance, in relation to classical Liouville-integrable motion, performing a canonical transformation to the action-angle variables completely trivializes the dynamics and represents it through decoupled one-dimensional oscillations. While there is no correspondingly general picture for quantum systems, and indeed the very definition of 'quantum integrability' in full generality is elusive [18,19], the fact that the classical limit of integrable systems is simple in a concrete and general sense suggests that some form of simplicity should be inherent to quantum integrable systems as well. Questions of quantum evolution complexity and in particular the way it may capture the difference between integrable and chaotic systems have recently been approached in [20,21]. (We also mention in passing the interesting recent preprint [22], where similar comparisons between integrable and chaotic systems are made using other information-theoretic measures unrelated to Nielsen's complexity.)

In attempts to work with Nielsen's complexity in concrete examples, an essential difficulty comes into play. Even if one approximates the relevant dynamics using a finite-dimensional Hilbert space of dimension $D$, the group of unitaries $U(D)$ to which the evolution operator belongs is a manifold of dimension $D^2$ (this will translate to group manifolds whose dimension is of order a million or a billion for the practical considerations in this paper). Even for moderate values of $D$, locating (or even approximating) the optimal geodesic on a manifold of such dimension is a formidably demanding problem. Indeed, already for a manifold as simple as a two-dimensional elipsoid of revolution, up to 4 distinct geodesics may connect two chosen points [23], and their lengths must be compared to select the shortest one. It is natural to expect that these alternative geodesics will proliferate dramatically as the number of dimensions grows. More than that, group manifolds contain maximal tori (Cartan subgroups), and on a torus, there is a discrete infinity of geodesics connecting any two points (these families of

curves will in fact play a central role in our subsequent considerations). The 'curse of dimensionality' one encounters in handling Nielsen's complexity is in fact ironic: while the optimal geodesic, if found, provides the 'easiest' way to construct the unitary operator in question, finding this geodesic on a group manifold of high dimension in itself poses an insurmountable challenge.

With direct evaluation of Nielsen's complexity being out-of-reach save for some low- dimensional toy examples, one must look for alternative approaches. One such approach has been pursued in [20, 21] and relies on tracking down the *conjugate points* (or 'focal points' [24]) along the geodesic lines. These points are defined as the loci where small deformations of the geodesic exist that do not upset the geodesic equation at linear order in the deformation magnitude. It can be shown that once a geodesic crosses such a conjugate point, it is guaranteed that a shorter geodesic will appear that should replace the original one for the purpose of complexity evaluation [20, 21, 24]. The picture is then that complexity grows linearly along the evolution geodesic $e^{-iHt}$ until a conjugate point is encountered, and then saturates. An advantage of conjugate points is that they can be identified on the basis of local analysis in the neighborhood of the original geodesic, in contrast to the global minimization necessary to find the true optimal geodesic. In practice, this analysis is implemented in terms of the geodesic deviation equation, as in [20, 21], where properties of conjugate points have been related to the spectral data of the evolution Hamiltonian. An inherent limitation of the conjugate point analysis is that, while it is intuitively in accord with the picture of complexity growth saturation, there is no controlled relation between conjugate points and the complexity plateau. Indeed, while it is true in general that shorter geodesics emerge once the evolution advances past a conjugate point, local analysis cannot tell whether these geodesics will themselves continue to grow, and thus one cannot infer conclusively whether a plateau has been reached. Furthermore, shorter geodesics can emerge in a manner not involving conjugate points (a situation referred to as 'geodesic loops' in [20, 21]) and this may also happen before the first conjugate point has been reached, offering possible complexity saturation mechanisms unrelated to conjugate points.

In view of the inherent limitations of the conjugate point analysis, it seems attractive to complement it with alternative approximate treatments in search of a more complete perspective. Our goal is this paper is precisely to develop one possible approach along these lines. One can think of this approach as a sort of 'variational minimization.' While Nielsen's complexity is exactly given by the absolute minimum of the distance from the origin to the given unitary operator measured along all possible curves, minimizing the same distance over any concrete prescribed set of curves in the very least gives an upper bound on the true complexity. The concrete set of curves we shall use is a very simple infinite family of geodesics of the bi-invariant metric on the group manifold, already employed for more limited related purposes in [20, 21, 25]. How good is the bound we find? One concrete criterion by which the bound can be judged is whether it succeeds in distinguishing between evolution operators characterizing motions of qualitatively different types, as in integrable vs. chaotic systems. We shall see that the bound we devise tackles this task adequately by systematically assigning lower values to integrable systems. (We shall also discover a broader set of appealing quantities analytically related to our complexity bound that distinguish integrable and chaotic systems even more effectively.)

Minimization over the set of curves we propose amounts, for a system with a Hilbert space of dimension $D$, to minimizing a quadratic polynomial over a $D$-dimensional set of integers $\mathbb{Z}^D$, a form of optimization that may be called 'integer quadratic programming' in the context of applied mathematics (the usage of 'programming' in optimization theory is independent of the notion of computer programming). This minimization task is directly related to the well-known *closest vector problem* (CVP) and the associated simpler *shortest vector problem*

(SVP), most commonly studied in relation to *lattice-based cryptography* [26, 27]. Exact solution of these problems is known to be extremely computationally demanding, with available algorithms running in time exponential in $D$. This difficulty, in fact, underlies the usage of these problems in open key cryptography (more than that, they are currently investigated for their ability to withstand quantum computer attacks within the domain of postquantum cryptography [28]). Thus, while we have considerably simplified the original notion of geodesic complexity, the result still involves a form of minimization that is intractable within a brute force approach. Nonetheless, it turns out that a range of attractive algorithms exist that run in polynomial time and provide approximate solutions for SVP (the Lenstra-Lenstra-Lovász algorithm [29, 30]) and for CVP (the nearest plane or Babai algorithm [31, 32]). Suboptimal solutions found using these algorithms still supply rigorous upper bounds on the genuine geodesic complexity, and it is these solutions that we shall investigate and find useful in capturing the difference between integrable and chaotic dynamics.

It remains to specify the classes of systems that we shall use as a testbed for exploring our ideas. To avoid dealing with the difficulties in defining complexity for infinite-dimensional Hilbert spaces, it is natural to focus first on systems with finite-dimensional Hilbert spaces. As a starter, we shall revisit the fermionic systems with polynomial Hamiltonians that have already been used for exploring complexity-related issues in [20, 21, 25]. Such Hamiltonians can be traced back to random matrix approaches to nuclear physics [33–37], and some of them have received considerable attention within high-energy theory under the name of SYK models, following [38, 39]. Our application of the lattice optimization techniques to construct bounds on complexity growth proves an immediate success for this class of models. Besides reproducing the general picture of complexity growth saturation already discussed in [20, 21, 25], for the 'integrable' case previously treated in [21] our methods automatically recover a considerably tighter upper bound on the complexity curve than the one derived in [21].

While discrete fermionic systems are very convenient for our studies, since they come by construction with finite-dimensional Hilbert spaces (this attractive feature is shared by spin chains, which we do not consider here), this convenience is not without a catch. One problem is that such systems do not have classical limits.[1] Since no universal definition of quantum integrability exists [18, 19], an explicit classical limit and the option of connecting to sharply defined classical notions of integrability would certainly offer an advantage for studying foundational features of the corresponding dynamics.

To bypass the lack of conventional classical limits for spin chains and fermionic systems, we proceed further with exploring our complexity bounds, and focus our attention on a special class of bosonic systems of considerable appeal for our studies. These *quantum resonant systems*, introduced and analyzed in [41], are described by Hamiltonians quartic in the creation-annihilation operators. Quantum resonant systems are attractive in that they combine the advantages of two classes of systems most commonly studied in relation to quantum chaos topics [42, 43]: spin chains and billiards. Just like billiards, they possess well-defined classical limits in the form of conventional classical Hamiltonian dynamics. Just like spin chains, their Hilbert spaces are (effectively) finite-dimensional and their quantum dynamics is solved by diagonalization of finite-sized matrices. (More precisely, the Hamiltonian is block-diagonal in the full infinite-dimensional Hilbert space, and all the blocks are of finite sizes; as a result, the dynamics separates block-by-block, and the Hilbert space becomes effectively finite-dimensional.)

---

[1]It is possible to reach a classical regime for spin chains by increasing the magnitudes of individual spins, as in [40]. This, however, differs from the conventional classical limit taken for physical states without varying the definition of the degrees of freedom.

Explicitly, quantum resonant systems are defined by the Hamiltonian

$$H = \frac{1}{2} \sum_{\substack{n,m,k,l=0, \\ n+m=k+l}}^{\infty} C_{nmkl}\, a_n^\dagger a_m^\dagger a_k a_l \,, \tag{1.1}$$

where $a_n$ and $a_n^\dagger$ with integer $n \geq 0$ are the usual bosonic creation-annihilation operators satisfying $[a_n, a_m^\dagger] = \delta_{nm}$. The interaction coefficients $C_{nmkl}$ are numbers that physically encode the strength of couplings between different bosonic modes and play a crucial role in defining the dynamics (integrable cases are given precisely by assigning very special values to these numbers). The name 'resonant' comes from the resonance condition $n+m = k+l$ imposed on the summation. This resonance condition is crucial for simplifying the diagonalization of the Hamiltonian (1.1) in a way that makes it easy to access its eigenvalues and eigenvectors.

The Hamiltonian (1.1) may appear not-too-familiar to many readers, but, as a matter of fact, the corresponding classical Hamiltonian system frequently arises as a controlled approximation to weakly nonlinear partial differential equations (PDEs) in strongly resonant domains, originating from a number of branches of physics and mathematics. Specifically, classical systems corresponding to (1.1), together with some closely related variations, have been studied in the following contexts: gravitational dynamics in anti-de Sitter (AdS) spacetimes [44–51] (typically, motivated by the AdS instability conjecture [52,53]); related dynamical problems for classical relativistic fields [54–60]; nonrelativistic nonlinear Schrödinger equations describing, among other things, the dynamics of Bose-Einstein condensates in harmonic potentials [61–70]; and integrable models for turbulence [71–77]. These classical systems display, for different choices of $C_{nmkl}$, a wide range of analytic and dynamical patterns ranging from full solvability [71–74] to Lax-integrability [71–77], partial solvability [54–58,64–66,78,79], turbulent cascades [47,71–77], as well as generic chaotic dynamics expected from a nonlinear system with an infinite number of degrees of freedom when the mode couplings $C_{nmkl}$ are chosen randomly. A recent review can be found in [80]. In contrast to the rich array of classical dynamical behaviors, the corresponding quantum theory is very economical in its structure and can be explored via an operation as simple as diagonalizing finite-sized numerical matrices [41]. (We mention in addition that (1.1) arises directly in the process of applying the standard Hamiltonian perturbation theory for the degenerate spectrum of quantum fields in strongly resonant domains at first order in the quartic interaction strength [81–84].)

Our exposition is organized as follows. In section 2, we review the geometric basics on the manifold of unitary operators, different possible metrics, geodesics, and our proposal for an upper bound on Nielsen's complexity. In section 3, we review the approximate lattice optimization techniques useful for estimating our bound, and put forward a concrete algorithmic implementation of these estimates. In section 4, we apply this construction to the case of SYK models previously studied from a similar perspective in [21]. We observe that a blind application of our methods that does not rely on any explicit analytic information about the models recovers and improves the bounds constructed in [21] by system-specific methods. We then proceed with the application of our techniques to bosonic quantum resonant systems, which display properties closer to what one would expect from generic integrable systems than what is seen in integrable SYK models, and start by reviewing the basic properties of this class of systems in section 5. Thereafter, in section 6, we present our main results in relation to the behavior of complexity in integrable and generic (chaotic) quantum resonant systems. We conclude with a summary and discussion of open problems.

## 2 An upper bound on geodesic lengths

The practicability of Nielsen's definition of complexity in quantum systems requires an effective way to handle the geodesics on the manifold of unitary operators. Although the idea to search for the shortest path connecting the identity operator with the time-dependent evolution operator and computing its length is very intuitive, it is forbiddingly hard to solve this problem in practice, even for reasonably small Hilbert space dimensions. Ultimately, one would like to probe thermodynamic and semiclassical limits, which requires techniques that are applicable to large Hilbert spaces.

Since the optimization problem involved in the computation of complexity seems *a priori* intractable, a natural approach is to try and simplify it by restricting the minimization to a prescribed family of curves. Such a procedure would certainly provide an upper bound on the complexity, and the key question is in finding a compromise between having a tractable minimization problem and having it produce a nontrivial, useful upper bound. The success of our approach is thus judged in terms of the existence of an efficient set of tools to perform the minimization procedure, and in terms of the ability of the resulting upper bound on Nielsen's complexity to distinguish between quantum systems with different properties.

The definition of geodesics and their corresponding lengths depends on the notion of distance one introduces on the manifold of unitary operators. We shall therefore start by describing some possible definitions for this metric. After reviewing the standard bi-invariant metric, whose associated geodesics will be a key ingredient in our subsequent analysis, we consider complexity metrics that generalize the bi-invariant metric by the introduction of penalty factors for a chosen set of 'nonlocal' directions. These latter metrics are what we shall actually use to measure the lengths in our variational minimization approach.

### 2.1 Group metrics and their geodesics

Consider the group of unitary operators $U(\mathcal{H})$ on a finite-dimensional Hilbert space $\mathcal{H}$, where we denote by $T_i$ the Hermitian generators of the associated Lie algebra, which we assume to be orthonormalized as

$$\mathrm{Tr}[T_i T_j] = \delta_{ij}. \tag{2.1}$$

The usual bi-invariant metric on the unitary group, defining the distance between two unitary operators $U$ and $U + dU$, is given by

$$ds^2_{\text{bi-inv}} = \mathrm{Tr}[dU^\dagger dU], \tag{2.2}$$

which is invariant under the multiplication of elements of $U(\mathcal{H})$ by any fixed unitary operator either on the left or on the right (hence the term 'bi-invariant').

The complexity of a unitary operator $U_{\text{target}}$ was defined by Nielsen as the length of the shortest curve connecting it to the identity [14]. Given a continuous path in $U(\mathcal{H})$ with boundary conditions

$$U(0) = \mathbf{I}, \qquad U(t) = U_{\text{target}}, \tag{2.3}$$

and velocity $V(t)$ defined by

$$\frac{dU(t)}{dt} = -iV(t)U(t), \qquad V = i\frac{dU}{dt}U^\dagger, \tag{2.4}$$

the associated path length in the bi-invariant metric is given by $\int_0^t dt' \left(\mathrm{Tr}[V(t')^2]\right)^{1/2}$. Then, the complexity in the bi-invariant metric of the unitary $U_{\text{target}}$ is obtained by minimizing this

length over all paths (2.4) subject to the boundary conditions (2.3):

$$\mathcal{C}_{\text{bi-inv}}(t) = \min \int_0^t dt' \big( \text{Tr}[V(t')^2] \big)^{1/2} . \tag{2.5}$$

This definition of complexity, however, does not take into account that some quantum operations may be easier to implement than others. (Furthermore, as we shall see below, it is too 'universal' and fails to distinguish effectively between different types of quantum motion.) In practice, easy operations are understood to be local or few-body operators, and a good complexity measure should be sensitive to this physical input. One way to guide the minimal length curves mostly through the local directions on the manifold of unitaries is to separate the tangent space into 'easy' and 'hard' directions by assigning a large penalty factor in the distance measure to the latter. We shall denote the corresponding easy and hard generators as $T_\alpha$ and $T_{\dot\alpha}$ respectively. The penalty factor may be chosen in proportion with the difficulty to implement the operation, and a variety of 'penalty plans' can be designed [15, 16]. In our treatment, we shall rely on a simple separation of the generators into two groups and assign a fixed penalty to all generators in the second group. We thus write

$$V = V_e + V_h \quad \text{with} \quad V_e \equiv V^\alpha T_\alpha , \quad V_h \equiv V^{\dot\alpha} T_{\dot\alpha} , \tag{2.6}$$

to define a new distance [12–14]

$$\mathcal{C}(t) = \min \int_0^t dt' \Big[ \text{Tr}[V_e(t')^2] + \mu \, \text{Tr}[V_h(t')^2] \Big]^{1/2} = \min \int_0^t dt' \Big[ \sum_\alpha (V^\alpha)^2 + \mu \sum_{\dot\alpha} (V^{\dot\alpha})^2 \Big]^{1/2} , \tag{2.7}$$

with a *cost factor* $\mu$ that is typically taken to be of the order of the dimension of the Hilbert space $\mathcal{H}$ [20, 21] or larger [12–14]. We shall denote this dimension $D$ so that the unitary evolution operators are $D \times D$ matrices, and focus on the assignment

$$\mu = D . \tag{2.8}$$

(Statements in the literature are often given in terms of the 'entropy' $S \equiv \log D$, but we shall not use this language.)

The notion of distance employed in (2.7) can be understood as a generalization of the bi-invariant metric (2.2). To see this, we first write (2.2) as

$$ds^2_{\text{bi-inv}} = \text{Tr}[dU^\dagger dU] = \sum_{kl} \text{Tr}[i \, dU U^\dagger T_k] \, \delta_{kl} \, \text{Tr}[i \, dU U^\dagger T_l] , \tag{2.9}$$

where we have used the completeness of the basis $T_i$. Then, the *complexity metric* is obtained by replacing [7, 15] the Kronecker symbol in (2.9) with a more general matrix $g$ as

$$ds^2 = \sum_{kl} \text{Tr}[i \, dU U^\dagger T_k] \, g_{kl} \, \text{Tr}[i \, dU U^\dagger T_l] = dt^2 \Big[ \sum_\alpha \big( \text{Tr}[V T_\alpha] \big)^2 + \mu \sum_{\dot\alpha} \big( \text{Tr}[V T_{\dot\alpha}] \big)^2 \Big] , \tag{2.10}$$

where our specific choice for $g$ is

$$g = \begin{pmatrix} \delta_{\alpha\beta} & 0 \\ 0 & \mu \, \delta_{\dot\alpha\dot\beta} \end{pmatrix} . \tag{2.11}$$

The metric (2.10) is manifestly invariant under multiplication by unitary group elements from the right, since $dU U^\dagger$ has this property, whereas it is not left-invariant. At $\mu = 1$, the metric (2.10) and its associated complexity (2.7) reduce to their respective bi-invariant counterparts (2.2) and (2.5).

Different definitions of the metric on the manifold of unitaries lead to different solutions for geodesic curves, and hence different notions of complexity. While geodesics of the metric (2.10) are in general very complicated, the geodesics of the bi-invariant metric (at $\mu = 1$) are simply of the form $e^{iV(t-t_0)}$ with $V$ being a time-independent Hermitian matrix. Heuristically, these paths of constant velocity define geodesics of the metric (2.2) because no direction is preferred in the space of unitaries in the absence of a penalty factor. More precisely, it is straightforward to see that the paths in $U(\mathcal{H})$ that extremize the length functional

$$L = \int ds_{\text{bi-inv}} = \int \left[ \text{Tr}\left(\dot{U}^\dagger \dot{U}\right) \right]^{1/2} dt \,, \tag{2.12}$$

satisfy the geodesic equation

$$\frac{dV^i(t)}{dt} = 0 \,, \tag{2.13}$$

with $V(t) = V^i(t) T_i$ as defined in (2.4) for a path $U(t)$. Therefore, the geodesics of the bi-invariant metric (2.2) must be of the form $e^{iV(t-t_0)}$.

For complexity-related considerations, one is interested in geodesics connecting the identity to a target unitary $U_{\text{target}} = \exp(-iHt)$ at a chosen moment $t$, with $H$ being the physical Hamiltonian. For the bi-invariant metric, these geodesics are of the form $U(t') = e^{-iH't'}$ for a Hermitian matrix $H'$ with

$$e^{-iH't} = e^{-iHt} \,, \tag{2.14}$$

at the given value of $t$. In a generic situation where degeneracies are absent, this last relation implies that the Hamiltonian $H$ and the velocity vector $H'$ share the same set of eigenvectors, while the eigenvalues $E'_n$ of $H'$ differ from the eigenvalues $E_n$ of the Hamiltonian by $2\pi k_n/t$ with arbitrary integers $k_n$:

$$H' = \sum_n \left( E_n - \frac{2\pi}{t} k_n \right) |n\rangle\langle n| \,. \tag{2.15}$$

Here, $E_n$ and $|n\rangle$ are the energy eigenvalues and eigenvectors of $H$. Geodesics of the bi-invariant metric connecting the identity and $U_{\text{target}} = \exp(-iHt)$ are

$$U_{\mathbf{k}}(t') = \exp(-iH't') \,, \tag{2.16}$$

where $H'$ is given by (2.15) and is parametrized by a $D$-dimensional vector of integers $\mathbf{k}$ whose components are $k_n$.

A choice has to be made at this point regarding whether the geodesics can run within the full manifold of unitaries $U(\mathcal{H})$, or only within the submanifold of unitaries whose determinant is equal to 1. Since the difference between general unitaries and those with determinant 1 is in the overall phase, and the phase of the wavefunction is unphysical, we do not expect a significant difference between the two definitions. If one restricts the geodesic to the unit determinant submanifold, as in [20,21], the condition $\sum_n k_n = 0$ must be imposed in (2.15). In most of our treatments below, we shall, however, deal with curves on the full group of unitaries, since it makes the analysis more transparent. (The unit determinant condition may be straightforwardly incorporated into a somewhat more involved version of our treatment.)

When $\mu \neq 1$, the curves (2.16) in general no longer define geodesics. While constant velocity vectors $V^i$ that lie either entirely in local directions or entirely in nonlocal directions still solve the geodesic equation, a constant vector with both local and nonlocal components does not define a geodesic for the complexity metric. We shall, however, use precisely the minimal length over this family of curves, computed with the complexity metric (2.10), to construct our complexity bound.

## 2.2 Complexity in the bi-invariant metric

Since the complexity of quantum evolution depends on the choice of the unitary group metric, a natural starting point is to ask whether the complexity (2.5) associated to the simplest bi-invariant metric (2.9), for which all geodesics are known, is powerful enough to be sensitive to dynamical details. (The family of toroidal geodesics (2.16) of this bi-invariant metric has been considered in relation to the bi-invariant complexity of chaotic SYK models in [20, 25].)

Consider a time-independent Hamiltonian $H$ with energy eigenvalues $E_n$. Combining (2.5) with the family of toroidal bi-invariant geodesics with velocity (2.15), the complexity at time $t$ is obtained by the following minimization over the set of $D$ integers $k_n$:

$$\mathcal{C}(t) = t \, \min\Big(\sum_n E_n'^{\,2}\Big)^{1/2} = \left[\min_{k_n \in \mathbb{Z}} \sum_n \Big(E_n t - 2\pi k_n\Big)^2\right]^{1/2}. \tag{2.17}$$

In other words, the complexity is the distance between the vector $E_n t$ and the nearest point of the hypercubic lattice $2\pi\mathbb{Z}^D$. From this expression, one observes that, at early times, while all the values $E_n t$ remain within the interval $[-\pi, \pi]$, the shortest curve connecting the identity and the evolution operator $U_{target} = e^{-itH}$ is the path defined by the Hamiltonian itself ($k_n = 0$), which results in a linear complexity growth

$$\mathcal{C}(t) = t \left(\text{Tr}[H^2]\right)^{1/2} = t \left(\sum_n E_n^2\right)^{1/2}. \tag{2.18}$$

This linear behavior continues until $t$ reaches the value of $(|E_n^{\max}| \, \pi)^{-1}$, when a shorter geodesic emerges defined by assigning the value $\pm 1$ to the integer $k_n$ corresponding to the maximal eigenvalue $E_n^{\max}$.

In view of the universal linear growth of complexity at early times given by (2.18), it is convenient to adopt normalization conventions[2] where this growth is identical and given by $\mathcal{C}(t) = t$ for all physical systems. This amounts to imposing

$$\text{Tr}[H^2] = \sum_n E_n^2 = 1, \tag{2.19}$$

which can always be accomplished by energy rescaling. Since the effect of energy rescaling depends on the (unphysical) common shift of energies by a constant, it is wise to fix this ambiguity as well by imposing

$$\text{Tr}H = \sum_n E_n = 0. \tag{2.20}$$

Evidently, any Hamiltonian can be subject to a common shift and scaling of energy so as to satisfy both (2.19) and (2.20), and this is the normalization we shall assume throughout this article. The early-time complexity growth will thereby be fixed for all physical systems as

$$\mathcal{C}(t) = t. \tag{2.21}$$

At later times, one has to find a set of integers $k_n$ that bring $E_n'$ as close to zero as possible to minimize (2.17). This is done by setting $k_n = \lfloor \frac{E_n t}{2\pi} \rfloor$, where $\lfloor \rfloor$ denotes rounding to the nearest integer. As $t$ grows, $E_n'$ can be brought closer and closer to zero, resulting in saturation of the complexity curve following the initial linear growth. Geometrically, this saturation is hardly

---

[2]We comment on the relation with conventions of [12, 20, 21] in footnote 7 in section 4.2.

surprising, since groups of unitaries are compact and the maximal distance between any two points is bounded from above.[3]

The observed saturation height of the bi-invariant complexity and the amplitude of oscillations around that plateau can in fact be predicted analytically by computing the distribution of distances between a point in $\mathbb{R}^D$ and the nearest point of the hypercubic lattice $2\pi\mathbb{Z}^D$. We will assume that a generic point along the line $E_n t$ is as close to the nearest point of the lattice as a randomly chosen generic point in space would be (this distance, in fact, concentrates on a specific value at large $D$, as we shall see below). We thus assume that, for sufficiently late times, the values $E_n t - 2\pi\lfloor\frac{E_n t}{2\pi}\rfloor$ that enter the expression for the complexity (2.17) will be distributed uniformly on the interval $[-\pi, \pi]$. The average length of a $D$-dimensional vector with components drawn from this uniform distribution is

$$\mu_x = \frac{1}{\pi^D}\int_0^\pi dx_1 \int_0^\pi dx_2 \cdots \int_0^\pi dx_D \Big(\sum_{i=1}^D x_i^2\Big)^{1/2}, \qquad (2.22)$$

where we have cut the integration interval in half using the reflection symmetry. One also has the corresponding evident formula for the variance of the length. The integrals involved are a particular case of the so-called *box integrals*, which can be analyzed at large $D$ using the central limit theorem [87]. One starts by considering the mean and variance associated to the variable $w = x_i^2$ for $x_i$ uniformly distributed in the interval $[0, \pi]$: $\mu_w = \pi^2/3$ and $\sigma_w^2 = 4\pi^4/45$. Then, the variable $z = \sum_{i=1}^D x_i^2$ has the mean and variance $\mu_z = D\pi^2/3$ and $\sigma_z^2 = 4D\pi^4/45$. Finally, since $\sigma_z$ is much smaller than $\mu_z$, one straightforwardly extracts from this the mean and variance of $x = \sqrt{z}$ by Taylor-expanding $\sqrt{z}$ around $z = \mu_z$:

$$\mu_x = \sqrt{\mu_z} = \pi\sqrt{\frac{D}{3}}, \qquad \sigma_x^2 = \frac{\sigma_z^2}{4\mu_z} = \frac{\pi^2}{15}, \qquad (2.23)$$

which are the expected complexity plateau height and variance. Note that $\sigma_x$ is negligible compared to the height $\mu_x$ at large $D$.

The concentration result expressed by (2.23) suggests that the saturation of the bi-invariant complexity is highly universal and does not really depend on the specific features of the chosen Hamiltonian $H$. This property, which we shall now validate in more detail, makes the bi-invariant complexity of limited use for physical applications, as it is essentially blind to important dynamical distinctions.

A crucial capacity one may expect from a quantity sensitive to the dynamics is its ability to distinguish between integrable and chaotic motion. We shall now show that the bi-invariant complexity (2.17) largely fails with this assignment. First, note that the bi-invariant complexity defined by (2.17) only knows about the specific Hamiltonian it is applied to through its energy eigenvalues $E_n$, while being oblivious to its eigenvectors (this will change once we move away from the bi-invariant metric in all of the subsequent sections). Properties of energy spectra are a central topic of quantum chaos theory [42, 43] (we shall give a brief practical overview of these matters in section 5.3). Most importantly, the distribution of appropriately normalized distances between neighboring energy levels is expected to coincide for integrable systems with the distribution of distances between points randomly thrown on a line [88], and, for chaotic systems, between eigenvalues of random matrices [89]. We can therefore test the sensitivity

---

[3]At very late times, which are effectively unobservable except for very low-dimensional Hilbert spaces, one expects Poincaré recurrences as the evolution operator passes close to **I**, and hence the complexity curve will drop again to low values. This happens when all $E_n t/2\pi$ are sufficiently close to integers. The equidistribution theorem [85] guarantees that this will happen at some time with an arbitrary prescribed precision, though generically the time required is exponential in $D$. Poincaré recurrences in small numbers of dimensions have been tracked down numerically in [86] using some of the lattice optimization techniques that will play a key role in our treatment below.

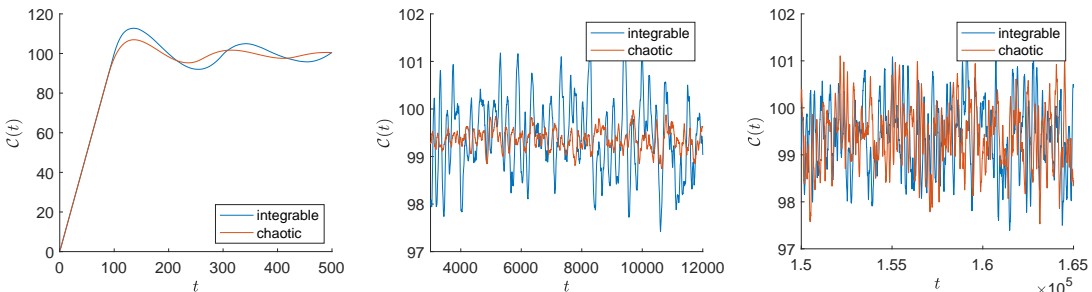

Figure 1: The bi-invariant complexity (2.17) for random energy spectra that mimic the energy level statistics expected from integrable and chaotic systems according to quantum chaos theory. The two curves are very similar, and the plateau properties are captured well by our estimates (2.23). The middle and right plots give an overview of the plateau at intermediate and late times, respectively. The numerically computed mean and variance of the two curves at late times are comparable and agree well with (2.23). A more subtle difference between the two curves is visible at intermediate times, where the chaotic curve momentarily displays a smaller vertical spread.

of (2.17) by feeding into it (a) random energy levels, and (b) eigenvalues of random matrices, and observing to what extent it can tell them apart.

Following these guidelines, as a representative 'integrable' spectrum $E_n$, we take a set of 3000 randomly chosen numbers drawn from a uniform distribution in the interval $[0, 1]$, while as a representative 'chaotic' spectrum, we take the eigenvalues of a randomly generated $3000 \times 3000$ Hermitian matrix with elements drawn from a normal distribution. The normalization (2.19-2.20) is enforced for these spectra via energy shift and scaling, in accord with our general conventions. The resulting complexity curves[4] are displayed in Fig. 1, and appear very similar. The linear growth is terminated at the same time for both models, resulting in identical plateau heights. Thus, at least at this level, (2.17) appears not powerful enough to distinguish between integrable and chaotic Hamiltonians. We have verified numerically that the plateau behavior agrees well with our estimate (2.23) in terms of both its height and variance. Moreover, we observe very similar complexity curves for integrable and chaotic spectra obtained using other distributions than the ones described above, which suggests that these properties are essentially universal. This confirms our intuition that the plateau statistics merely reflects the distribution of distances from random points in a $D$-dimensional Euclidean space to the nearest point of a hypercubic lattice.

Interestingly, while the variance about the mean saturation value, computed in a sufficiently late time window, agrees well numerically with the estimate (2.23) for both types of dynamics, the 'chaotic' curve in Fig. 1 displays less sharp spikes in the intermediate time regime compared to the integrable one. Thus, exploring some fine features of the plateau (and not just looking at the typical complexity values) may reveal differences between integrable and chaotic motion. This question would be rather interesting to investigate from the mathematical perspective, but it is far from our concrete objective to develop effective complexity measures. We additionally mention in passing that another interesting complexity-inspired quantity, not directly related to Nielsen's definition, has been proposed in [90] under the name of 'spectral complexity.' This quantity is also expressed through the energy eigenvalues and not eigenvectors, but is expected to distinguish between integrable and chaotic spectra.

---

[4]The data and code used for the numerics in this article are available in the Zenodo data repository at https://doi.org/10.5281/zenodo.6339975.

## 2.3 A variational bound on complexity with nonlocality penalties

With the bi-invariant complexity displaying a broad lack of sensitivity to the actual dynamics of the physical system under consideration, we must turn to more refined complexity measures, and the definition (2.7) that introduces the nonlocality penalty $\mu$ for 'hard' directions on the manifold of unitaries is a good starting point. An issue with this definition is that the problem of finding shortest geodesics of the metric (2.10) is expected to be intractable for Hilbert spaces with any sizable number of dimensions.

The question is then with finding a compromise between simplifying the definition (2.7), while ensuring that this simplified definition is sensitive to physical features of interest (integrability and chaos, for example). For one thing, driving this simplification as far as removing the distinction between easy and hard directions and returning to the bi-invariant metric would not work, as we have already explained.

One way to simplify a minimization problem is to restrict the minimization to a subset of the original set of configurations, in our case to a subset of curves. Minimizing over such a subset evidently provides an upper bound on the full minimization. It is in this specific sense that we use the term 'variational' (akin to variational methods in quantum mechanics, where upper bounds on the ground state energy are obtained by minimizing the Hamiltonian over a restricted set of wavefunctions). Which subset of curves should we choose? We find the infinite set of 'toroidal' curves (2.15-2.16) attractive in this regard. These curves are geodesics of the bi-invariant metric, though they are in general no longer geodesics once the penalty factor has been introduced. They still provide a valid basis for restricted minimization, and they explore infinitely many ways to connect a given unitary operator to the identity. As simple-minded as this approach is, we shall see that it works in practice, and produces an upper bound on complexity that is sensitive to the physics that is being probed. This bound and its sensitivity are the key messages of our present treatment.

Once we have replaced the minimization over all curves in (2.7) with minimization over the curves (2.16), we obtain an optimization problem over a set of $D$ integers $k_n$ that parametrize these curves. It turns out that this integer optimization problem has been widely studied. Its exact solution is still inaccessible for any sizable Hilbert spaces (and this difficulty underlies the usage of these problems in lattice-based cryptography), but a family of powerful algorithms exist that effectively generate approximate solutions to this optimization problem, and they will turn out sufficient for our purposes. All of this will be covered in detail in the next section.

Coming back to the concrete implementation of our restricted minimization, we evaluate the complexity (2.7) with cost factor $\mu$ on the family of curves (2.16). With $E_n$ and $|n\rangle$ being the eigenvalues and eigenvectors of the Hamiltonian $H$, we first express the operators $|n\rangle\langle n|$ in terms of local and nonlocal generators ($T_\alpha$ and $T_{\dot\alpha}$ respectively),

$$|n\rangle\langle n| = c_n^\alpha T_\alpha + c_n^{\dot\alpha} T_{\dot\alpha}. \tag{2.24}$$

The cost factor $\mu$ can then be introduced into the norm of $H'$:

$$||H'||_\mu^2 = \sum_\alpha \Big[ \sum_n \Big(E_n - \frac{2\pi k_n}{t}\Big)c_n^\alpha \Big]^2 + \mu \sum_{\dot\alpha} \Big[ \sum_n \Big(E_n - \frac{2\pi k_n}{t}\Big)c_n^{\dot\alpha} \Big]^2. \tag{2.25}$$

In order to rewrite this expression in a more structured form, we define the matrix

$$Q_{mn} \equiv \sum_{\dot\alpha} \langle n|T_{\dot\alpha}|n\rangle\langle m|T_{\dot\alpha}|m\rangle = \delta_{mn} - \sum_\alpha \langle n|T_\alpha|n\rangle\langle m|T_\alpha|m\rangle, \tag{2.26}$$

where the sum over $\dot\alpha$ runs over the nonlocal directions, and the sum over $\alpha$, over all local directions (this second representation is often more practical computationally). Note that **Q**

is manifestly nonnegative, with eigenvalues between 0 and 1. Indeed, for any vector $y_m$,

$$\sum_{mn} Q_{mn} y_m y_n = \sum_{\dot{\alpha}} \left( \sum_n \langle n | T_{\dot{\alpha}} | n \rangle y_n \right)^2, \tag{2.27}$$

which shows that the eigenvalues of $\mathbf{Q}$ are nonnegative if one takes $y$ to be an eigenvector. Similarly, from the second equality in (2.26) it follows that the eigenvalues cannot exceed 1.

Importantly, the vector consisting of the energy eigenvalues $E_n$ is a null vector of $Q$ since the Hamiltonian is purely local:

$$\sum_{m=1}^{D} Q_{nm} E_m = \sum_{\dot{\alpha}} \sum_{m=1}^{D} \langle n | T_{\dot{\alpha}} | n \rangle E_m \langle m | T_{\dot{\alpha}} | m \rangle = \sum_{\dot{\alpha}} \langle n | T_{\dot{\alpha}} | n \rangle \mathrm{Tr}(T_{\dot{\alpha}} H) = 0 . \tag{2.28}$$

In general, if one increases the number of generators declared to be local (easy), the $Q$-matrix moves from the identity (all generators are nonlocal) to zero (all generators are local). We shall see that the way this transition happens differs significantly depending on the type of physical systems one studies.

With the $Q$-matrix, we can recast (2.25) as

$$\|H'\|_{\mu}^2 = \sum_{mn} \left( E_n - \frac{2\pi k_n}{t} \right) \left( \delta_{nm} + (\mu - 1) Q_{nm} \right) \left( E_m - \frac{2\pi k_m}{t} \right) . \tag{2.29}$$

As a consequence, minimizing the complexity over the family of paths (2.16) with velocity (2.15) produces the following upper bound on complexity

$$\mathcal{C}_{\mathrm{bound}}(t) = \min_{\mathbf{k} \in \mathbb{Z}^D} \left\{ \sum_{mn} (E_n t - 2\pi k_n) \left[ \delta_{nm} + (\mu - 1) Q_{nm} \right] (E_m t - 2\pi k_m) \right\}^{1/2} . \tag{2.30}$$

This bound will be the main object of our study. We shall discuss in the next section how to implement the integer minimization appearing in (2.30) in practice.

We remark that the $Q$-matrix is structurally rather similar to the $R$-matrix introduced in [20] while studying conjugate points along the geodesics of the metric (2.10) and given by

$$R_{mn} = \frac{\sum_{\alpha} |\langle m | T_{\alpha} | n \rangle|^2}{\sum_{\alpha} |\langle m | T_{\alpha} | n \rangle|^2 + \sum_{\dot{\alpha}} |\langle m | T_{\dot{\alpha}} | n \rangle|^2} . \tag{2.31}$$

The denominator comes from the restriction to the submanifods of unitaries with determinant 1 in [20], while the numerator is similar to the definition of $\delta_{mn} - Q_{mn}$, with a slightly different combination of the generators and eigenvectors. The mathematical properties resulting from these two definitions, and in particular the eigenvalue spectra that will play a considerable role below are, however, rather different for these two related matrices.

## 3 Lattice optimization

The complexity bound (2.30) can be understood geometrically as the distance between the vector $E_n t$ in $\mathbb{R}^D$ and the nearest point of the hypercubic lattice $2\pi\mathbb{Z}^D$, with distances in $\mathbb{R}^D$ measured using the metric $\mathbf{I} + (\mu - 1)\mathbf{Q}$. When $\mu = 1$, one returns to the bi-invariant case, where the $Q$-term drops out and the metric becomes Euclidean. (The lattice is evidently orthogonal with respect to the Euclidean metric, but not with respect to the metric involving a generic $\mathbf{Q}$.) It turns out that this form of integer optimization has been widely studied under the name of the *closest vector problem* (CVP). The main goal of this section is to review this

problem, together with the closely related *shortest vector problem* (SVP), in a manner adapted to evaluating (2.30).

Finding the nearest lattice point is easy on an orthogonal lattice, and that is precisely what we did in section 2.2: one must simply expand the given vector in the lattice basis and round each component to the nearest integer; the resulting integer vector will specify the nearest lattice point. This simple recipe no longer works, however, if the basis is not orthogonal. In fact, it is extremely difficult to find the nearest point of a generic lattice when the number of dimensions $D$ is large: the existing algorithms for solving this problem exactly run in time exponential in $D$, and it is believed that no polynomial-time algorithms exist. (A much deeper and systematic account from the standpoint of computational complexity can be found in [91].)

In fact, the difficulty of exactly solving lattice optimization problems (including CVP) is precisely what underlies their usage in the field of lattice-based cryptography [26, 27, 91]. An early foundational proposal for a cryptographic system functioning along these lines is due to Goldreich-Goldwasser-Halevi [92], and it revolves entirely around applications of CVP. (This cryptographic scheme has been later subject to critique [93], which has led to further developments [91].) In this protocol, there is a basis in which performing CVP is easy (for example, an orthogonal basis), known only to the receiver, and a generic non-orthogonal basis of the same lattice distributed publicly. A message is transcribed into a lattice point specification and then shifted by a small error before transmission. Since finding the closest lattice point to the transmitted signal (removing the error and recovering the message) is easy in the recipient's secret basis, the message can be successfully decoded by its addressee. But both solving CVP in the publicly known non-orthogonal basis or recovering the orthogonal basis from the non-orthogonal one are believed to be exponentially difficult in the number of dimensions D, and hence unfeasible to the public. Curiously, the minimization procedure necessary for exact evaluation of our complexity bound (2.30) is precisely the decoding step of the Goldreich-Goldwasser-Halevi cryptographic protocol.

At this point, the situation may appear desperate. While we have traded the search for the shortest geodesic on a $D^2$-dimensional manifold of unitaries for the much simpler minimization problem (2.30) over a $D$-dimensional set of integers, this new minimization problem is still complex enough to be used for constructing cryptographic 'trapdoor' functions, and hence, for all practical purposes, unsolvable. Unlike cryptographers, however, we are not, strictly speaking, concerned with implementing the minimization (2.30) exactly. It would suffice to find a solution that is suboptimal, but close enough to the true minimum to reflect interesting information. What we intend to report is precisely a successful construction of the sort. (Evidently, any upper bound on (2.30) is still a valid upper bound on the true Nielsen's complexity, and the specific upper bound we shall construct will turn out to reflect sensitively the kind of physical system one studies.)

While algorithms for exactly solving SVP and CVP run in time exponential in $D$, there are effective polynomial-time algorithms that typically provide good, though suboptimal, solutions. For SVP, this is the Lenstra-Lenstra-Lovász (LLL) basis reduction algorithm [29, 30] (and further related improvements). We shall review it first since this basis reduction is normally performed before attempting to solve CVP. (We will comment later on the relative importance of the different algorithms involved.) Then, for CVP, one has the Babai nearest plane algorithm [31, 32], which provides a significant improvement for non-orthogonal lattices over the naive rounding we had used in section 2.2.

We shall proceed shortly with a review of the relevant lattice optimization methods. However, these methods are customarily presented in a format where the ambient space metric is Euclidean but the lattice is arbitrary. In our formulation of (2.30), the ambient metric is $\mathbf{I} + (\mu - 1)\mathbf{Q}$ while the lattice is hypercubic. Of course, it is straightforward to relate the two

setups via a linear transformation, which we shall now do explicitly. We first diagonalize $\mathbf{Q}$ using a unitary matrix $\mathbf{V}$ as $\mathbf{Q} = \mathbf{V}^\dagger \mathbf{D} \mathbf{V}$ and then define the new matrix $\tilde{\mathbf{V}} = (\mathbf{I} + (\mu - 1)\mathbf{D})^{1/2}\mathbf{V}$. Then, (2.30) is rewritten in the form

$$\mathcal{C}_{\text{bound}}(t) = 2\pi \left\{ \min_{\tilde{\mathbf{k}} \in \mathcal{L}} \left\| \frac{\tilde{\mathbf{E}}t}{2\pi} - \tilde{\mathbf{k}} \right\|^2 \right\}^{1/2} . \tag{3.1}$$

Here, $\| \cdot \|$ denotes the Euclidean norm, while $\tilde{\mathbf{E}} \equiv \tilde{\mathbf{V}}\mathbf{E}$, and $\tilde{\mathbf{k}} \equiv \tilde{\mathbf{V}}\mathbf{k}$ is a vector in the lattice $\mathcal{L}$ generated by integer combinations of the basis vectors $\{b_1, \ldots, b_D\} = \{\tilde{\mathbf{V}}e_1, \ldots, \tilde{\mathbf{V}}e_D\}$, where $e_1 = (1, 0, 0, \ldots), \ldots, e_D = (0, \ldots, 0, 1)$ is the standard hypercubic basis. We have thus recast the complexity bound computation in the canonical CVP form (3.1) where one must simply find the point of the lattice $\mathcal{L}$ closest to the vector $\tilde{\mathbf{E}}t/2\pi$.

We shall then proceed to review the available CVP techniques for a general lattice generated by a given set of basis vectors $\{b_1, \ldots, b_D\}$. As we have already remarked, CVP is trivial if the basis is orthogonal and reduces to rounding to the nearest integers. It is then natural that one should start by redefining the lattice basis to make it as orthogonal as possible (as well as attempting to make the basis vectors short). This process is known as *basis reduction* and it is commonly discussed in relation to the shortest vector problem (finding the shortest integer combination of the basis vectors). We shall therefore review basis reduction first.

## 3.1 Basis reduction

A lattice $\mathcal{L}$ generated by a (full rank) basis $\{b_1, \ldots, b_D \in \mathbb{R}^D\}$ is defined as the set of points

$$\mathcal{L}(b_1, \ldots, b_D) = \left\{ \sum_{i=1}^{D} k_i b_i, k_i \in \mathbb{Z} \right\} . \tag{3.2}$$

The basis of a given lattice is evidently not unique, and linear combinations of $b_i$ with integer coefficients exist that generate exactly the same lattice. The choice of a good lattice basis is central for discussions of lattice optimization problems.

A natural question that often occurs in optimization applications is to find the shortest vector $\lambda$ on the lattice. For 2-dimensional lattices, this problem was solved exactly by Gauss in the XIX century. In higher dimensions, it is widely believed that no polynomial-time algorithm (with respect to $D$) that finds the exact solution exists. A systematic review can be found in [27, 30, 32].

Useful approximate solutions of the shortest vector problem can however be found using the celebrated Lenstra-Lenstra-Lovász (LLL) algorithm [29] that runs in polynomial time. This algorithm introduces the notion of *basis reduction* that will be used in our subsequent discussion of CVP. The idea is to start with the lattice basis $B$ (represented as a matrix containing the basis vectors as its columns)[5] and define a new basis $B'$ with shorter basis vectors. Since the volume of the unit cell defined by a basis is unaltered by a change of basis, the lengths of the basis vectors are the smallest when the vectors are as orthogonal as possible. Therefore, the aim of a basis reduction algorithm is to work towards a new set of basis vectors, obtained by considering integer linear combinations of the original set, where the unit cell has been reshaped to be rounder and less elongated. The compromise is between improving the basis as much as possible while still having an algorithm that runs within an acceptable time frame. The LLL algorithm does a good job in managing this compromise.

---

[5] Specifically, we denote by $B$ the $D \times D$ matrix containing as its columns the basis vectors $b_1, \ldots, b_D$.

Before delving into the details of the LLL algorithm, we recall that the orthogonality properties of a set of vectors can be quantified by considering the Gram-Schmidt procedure

$$b_i^* \equiv b_i - \sum_{j<i} \mu_{ij} b_j^* \quad \text{with} \quad \mu_{ij} = \frac{\langle b_i, b_j^* \rangle}{\langle b_j^*, b_j^* \rangle}, \tag{3.3}$$

where the orthogonalized basis vectors are denoted by $b_i^*$, and $\langle \cdot, \cdot \rangle$ is the Euclidean inner product. Geometrically, the procedure computes $b_i^*$ at each step by subtracting from $b_i$ its orthogonal projection onto the subspace spanned by the subset of basis vectors $\{b_1^*, \ldots, b_{i-1}^*\}$ (or equivalently $\{b_1, \ldots, b_{i-1}\}$). The coefficients $\mu_{ij}$ are called Gram-Schmidt coefficients and the terms $\mu_{ij} b_j^*$ are the orthogonal projections of $b_i$ onto $b_j^*$. One can visualize the relation between the two bases by expressing the basis $\{b_1, \ldots, b_D\}$ in the basis of normalized Gram-Schmidt vectors $\{b_1^*/||b_1^*||, \ldots, b_D^*/||b_D^*||\}$ using the matrix

$$\begin{pmatrix} ||b_1^*|| & \mu_{2,1}||b_1^*|| & \mu_{3,1}||b_1^*|| & \cdots & \mu_{D,1}||b_1^*|| \\ 0 & ||b_2^*|| & \mu_{3,2}||b_2^*|| & \cdots & \mu_{D,2}||b_2^*|| \\ 0 & 0 & ||b_3^*|| & \cdots & \mu_{D,3}||b_3^*|| \\ \vdots & \vdots & \vdots & \vdots & \ddots \\ 0 & 0 & 0 & \cdots & ||b_D^*|| \end{pmatrix}. \tag{3.4}$$

In practice, this matrix can be obtained by considering the orthogonal-triangular decomposition of the matrix of basis vectors $B = OR$, where $O$ is an orthogonal matrix whose columns are the normalized Gram-Schmidt vectors and $R$ is given by the upper triangular matrix written above.

Given the associated Gram-Schmidt vectors (3.3), a basis $\{b_1, \ldots, b_D\}$ is defined to be *LLL-reduced* if it satisfies the following two conditions:

- $|\mu_{ij}| \leq \frac{1}{2}$. This condition is usually referred to as *size reduction*. It ensures that the basis vectors $b_i$ are close to being mutually orthogonal. This can be seen from (3.4), where the condition puts a bound on the off-diagonal terms.

- $\delta ||b_i^*||^2 \leq ||b_{i+1}^*||^2 + \mu_{i+1,i}^2 ||b_i^*||^2$ for all $i$ and a given $\delta \in (1/4, 1]$. This condition is called *Lovász condition*. The bigger the chosen value for $\delta$, the stronger the condition. It imposes a bound on the decay of the lengths of the Gram-Schmidt basis vectors $||b_i^*||$ since for every $i$ we have

$$\left( \delta - \mu_{i+1,i}^2 \right)^{1/2} ||b_i^*|| \leq ||b_{i+1}^*||,$$

with $|\mu_{i+1,i}| \leq \frac{1}{2}$ by the size reduction condition.

The LLL algorithm then finds a basis satisfying these two conditions. The algorithm is given as its input an arbitrary basis $B$, and consists of the following three steps:

1. Compute the Gram-Schmidt orthogonalized basis $B^*$ and the Gram-Schmidt coefficients $\mu_{ij}$.

2. *Size reduce $B$*. The goal of this step is to shift the Gram-Schmidt coefficients $\mu_{ij}$ by integers such that they obey the size reduction condition $|\mu_{ij}| \leq \frac{1}{2}$. For a given basis vector with index $i$, it sets $b_i \leftarrow b_i - \lfloor \mu_{ij} \rceil b_j$ going from $j = i - 1$ down to $j = 1$. The coefficients $\mu_{ij}$ may change at every iteration and need to be updated accordingly. Note also that while this procedure changes the basis vectors $b_i$ and the coefficients $\mu_{ij}$, it does not change the Gram-Schmidt basis vectors $b_i^*$ since the component of $b_i$ perpendicular to $\{b_1, \cdots, b_{i-1}\}$ is not modified by the size reduction procedure.

3. If the Lovász condition $\delta||b_{i-1}^*||^2 \leq ||b_i^*||^2 + \mu_{i,i-1}^2||b_{i-1}^*||^2$ is violated for a given $i$, swap $b_{i-1}$ and $b_i$.

These three steps are performed for every index $i$ starting at $i = 2$, where step 3 sets $i \to i+1$ if the Lovász condition holds and $i \to i-1$ otherwise, after swapping. The Lovász condition is designed to indicate when a size-reduced basis can be further improved after swapping neighboring basis vectors. While a small Gram-Schmidt coefficient $\mu_{i,i-1}$ can imply that the two associated basis vectors are close to orthogonal, it can also signal that the norm of $b_{i-1}$ is large and thereby suppresses $\mu_{i,i-1}$ while the angle between $b_i$ and $b_{i-1}$ is not necessarily close to $\pi/2$. In such a situation, it might be beneficial to swap the order of $b_i$ and $b_{i-1}$ which may result in an updated Gram-Schmidt coefficient larger than $1/2$ allowing for further size reduction. For the implementation of the LLL procedure in MATLAB, we found the code provided with [94] very useful.

Note that after the Gram-Schmidt basis and coefficients have been computed in step 1, in the subsequent iterations it is not necessary to perform the orthogonalization procedure on the whole basis after every swap in step 3. After a swap has occurred, the matrix (3.4) will acquire a non-zero entry in the lower triangular part with indices $(i, i-1)$. To set this entry to zero again, one can perform a rotation in the two-dimensional plane spanned by the directions $i-1$ and $i$ in the matrix (3.4), sometimes referred to as a Givens rotation.

A basis obtained in this way contains as its first basis vector $b_1$ a vector that is relatively short and provides an approximation to the shortest vector in a lattice. To see this, first note that the two conditions above imply

$$||b_1^*||^2 \leq (\delta - \tfrac{1}{4})^{-(i-1)}||b_i^*||^2 \leq (\delta - \tfrac{1}{4})^{-(D-1)}||b_i^*||^2 \,, \tag{3.5}$$

for any $i$, where we have repeatedly used $||b_i^*||^2 \leq (\delta - \tfrac{1}{4})^{-1}||b_{i+1}^*||^2$ to obtain the first inequality, and then $(\delta - \tfrac{1}{4})^{-i} \leq (\delta - \tfrac{1}{4})^{-D}$ to obtain the second one. This implies the bound

$$||b_1|| \leq (\delta - \tfrac{1}{4})^{-(D-1)/2}\min||b_i^*|| \leq (\delta - \tfrac{1}{4})^{-(D-1)/2}||\lambda|| \,, \tag{3.6}$$

where $\lambda$ denotes the shortest vector in the lattice and use has been made of the inequality $||\lambda|| \geq \min_i||b_i^*||$ (proposition 3.14 in [32]).

Theoretical discussions of the LLL algorithm often revolve around the value $\delta = 3/4$. With this value, the LLL algorithm finds a lattice vector that is longer than the shortest lattice vector by a factor of $2^{(D-1)/2}$ in the worst case. The running time of the LLL algorithm on the other hand can be estimated to be of the order $\mathcal{O}(D^5 \log^2 \mathcal{B})$, where $\mathcal{B}$ is an upper bound on the lengths of the Gram-Schmidt basis vectors $||b_i^*||$. In practice, however, $\delta$ is often taken to be close to 1, as we will also be doing in our numerical work in the next sections.

While the errors exponential in $D$ present in the worst case bound may give the impression that the algorithm is useless, it is known to perform considerably better in practice than what the worst case bound may suggest. In [95], a series of numerical experiments was undertaken using randomly generated bases to assess the average performance of the LLL algorithm (and some of its improvements), and it was suggested that the typical error in length relative to the exact shortest vector is only $(1.02)^D$, which is tiny for large $D$ compared to the worst case bound $(4/3)^{(D-1)/2}$ when $\delta$ is close to 1.

## 3.2 The closest vector problem

Given a lattice $\mathcal{L}$ with basis $\{b_1, ..., b_D\}$ and a vector $x \in \mathbb{R}^D$, CVP is the problem of finding a point of the lattice $w \in \mathcal{L}$ that is closer to $x$ than any other lattice point. More generally, solving CVP within the approximation factor $\gamma \geq 1$ amounts to finding $w$ that satisfies

$$||w - x|| \leq \gamma||u - x||, \quad \forall u \in \mathcal{L}. \tag{3.7}$$

Evidently, $\gamma = 1$ corresponds to the exact CVP.

We can expand the vector $x$ in the lattice basis as

$$x = \sum_{i=1}^{D} c_i \, b_i \,, \tag{3.8}$$

with coefficients $c_i \in \mathbb{R}$. A simple approximation to the solution of CVP is then given by just rounding all coefficients to the nearest integer

$$w = \sum_{i=1}^{D} \lfloor c_i \rceil b_i \,. \tag{3.9}$$

This straightforward procedure was applied to the computation of the bi-invariant complexity in section 2.2 since for an orthogonal basis in the Euclidean norm this method is guaranteed to give the exact solution. However, using the complexity metric for computing the lengths on the manifold of unitaries results in an anisotropic metric $\mathbf{I} + (\mu - 1)\mathbf{Q}$ in the complexity bound (2.30), and hence in a nonorthogonal lattice $\mathcal{L}$ in the CVP problem (3.1). Under such circumstances, naive rounding is no longer efficient. (We shall in fact see in our numerical experiments that replacing naive rounding with a more optimal CVP technique is the most crucial step in recovering a useful complexity bound.) If the basis is LLL-reduced with $\delta = 3/4$, it was shown by Babai [31] that the naive rounding method (3.9) finds a lattice point $w$ closest to $x$ within an approximation factor of $\gamma = 1 + 2D(9/2)^{D/2}$. The same paper suggested a significantly better optimized approach.

An improvement over naive rounding can be achieved with Babai's *nearest plane algorithm* [31, 32], which is designed to deal with the skew of a lattice in an efficient way. Starting with an input (3.8) in $\mathbb{R}^D$, the essence of the method is that (instead of rounding the components) one first considers a family of $(D-1)$-dimensional lattice hyperplanes, picks the one closest to $x$, and descends orthogonally to that hyperplane. Then, one arrives at a $(D-1)$-dimensional CVP within that hyperplane, and applies the same technique recursively. Eventually, a 0-dimensional hyperplane will be reached, and that is simply a lattice point, which is declared to be the approximate CVP solution. Lattice hyperplanes are closely linked with the Gram-Schmidt vectors (3.4), and it is not surprising that these vectors will appear prominently in the technical implementation of the algorithm.

In practice, the Babai algorithm runs as follows. We start by searching the $(D-1)$-dimensional hyperplane

$$H_c^{(D-1)} = \left\{ \sum_{i=1}^{D-1} a_i b_i + c \, b_D \mid a_i \in \mathbb{R}, c \in \mathbb{Z} \right\} , \tag{3.10}$$

closest to $x$ (note that the actual closest lattice point to $x$ may be absent from the closest hyperplane in this family, but this does not prevent the Babai algorithm from finding a good approximate CVP solution). From the matrix representation of the basis vectors $b_i$ in their orthogonal decomposition (3.4), it follows that the vector $b_D^*$ is perpendicular to all the hyperplanes (3.10). Moreover, the decomposition

$$x = \sum_{i=1}^{D} l_i b_i^*, \qquad l_i \equiv \frac{\langle x, b_i^* \rangle}{\langle b_i^*, b_i^* \rangle} \,, \tag{3.11}$$

tells us that the closest hyperplane is (3.10) is the one with $c = \lfloor l_D \rceil$. We then construct the orthogonal projection $x_\perp$ of $x$ onto the hyperplane (3.10) with $c = \lfloor l_D \rceil$:

$$x = x_\perp + (l_D - \lfloor l_D \rceil)b_D^* \,. \tag{3.12}$$

Evidently, $x_\perp$ lies in the hyperplane (3.10) with $c = \lfloor l_D \rceil$, which also carries a $(D-1)$- dimensional sublattice of the original lattice defined by $D-1$ arbitrary integers $q_i$ as

$$\sum_{i=1}^{D-1} q_i b_i + \lfloor l_D \rceil b_D \,. \tag{3.13}$$

Continuing our attempt to solve CVP, it is natural to proceed looking for the point of this $(D-1)$-dimensional lattice closest to $x_\perp$. But this is exactly a $(D-1)$-dimensional CVP. Hence, we have reduced the number of dimensions by 1, and we can restart the algorithm described above recursively, $D$ times overall, whereupon we will reach a lattice point (0-dimensional hyperplane) that serves as the Babai approximation for CVP, as intended.

For a basis that is LLL-reduced with $\delta = 3/4$, the nearest plane algorithm finds a lattice point close to $x$ within approximation factor $\gamma = 2^{D/2}$. As with the LLL algorithm, the practical value of the Babai algorithm is that it typically performs considerably better than this worst case bound. While the algorithm is slower than the simpler rounding method (3.9), it still runs in polynomial time and does not represent a significant computational burden.

A few possible practical improvements over the basic implementation of the algorithm we have described above were listed in [92], though we shall not pursue them here. On the other hand, we have found some use in the steepest descent or 'greedy' algorithm that has been applied to related lattice optimization questions in [96]. Consider a basis $\{b_i | i = 1, \cdots, D\}$ for the lattice of interest, a target vector $x \in \mathbb{R}^D$ and a seed lattice vector $w$ that is the current best approximation to the solution of the CVP with target vector $x$. The idea of the greedy algorithm is to verify whether moving along one of the basis vectors $b_i$, by taking an integer number of steps $c$ in that direction, can reduce the distance to $x$. Consider

$$||w + c\, b_i - x||^2 - ||w - x||^2 = c\, g_i + ||b_i||^2 c^2 \,, \tag{3.14}$$

where

$$g_i \equiv \frac{\partial \left( ||w + c\, b_i - x||^2 \right)}{\partial c}\bigg|_{c=0} = 2\langle w - x, b_i \rangle \tag{3.15}$$

is the gradient of the distance function $||w + c\, b_i - x||^2$ at $c = 0$ in the direction of $b_i$. Minimization of (3.14) over integer values of $c$ is achieved by choosing

$$c = \lfloor -g_i/(2||b_i||^2) \rceil \,. \tag{3.16}$$

This value is nonzero when $|g_i| > ||b_i||^2$, signifying that one can get closer to $x$ by moving in that direction. The gain (3.14) corresponding to (3.16) can then be computed. The greedy algorithm consists in computing this gain in every direction $i = 1, \cdots, D$ of the lattice and moving in the direction of maximal gain by updating $w \to w + c^{opt} b_{i^{opt}}$. Thereafter, one should restart the process at the new value of $w$, and continue until no further gain is possible, that is, when all $c$'s given by (3.16) vanish.

The greedy algorithm depends, as do other algorithms we have discussed, on the choice of lattice basis. In view of the available bounds and existing practices discussed above, we will work with the LLL-reduced basis. We also note that although the greedy descent method does not, by itself, achieve a reduction of the distance of the naively rounded lattice point (3.9) comparable to the Babai nearest plane algorithm, we will observe in our numerical study that, when applied to the output of the Babai algorithm with directions $b_i$ defined by the LLL basis, the greedy algorithm does lead to noticeable improvements in some cases.

## 3.3 Complexity bounds from lattice optimization

We propose to tackle the problem of minimizing (2.30) over the lattice $2\pi\mathbb{Z}^D$ by making use of the lattice optimization techniques described above. In practice, one needs to start by diagonalizing the Hamiltonian of the system whose complexity we want to analyze. One specifies

a definition for the complexity basis, choosing the subset of local generators guided by physical principles. (In our applications, $k$-body operators with $k$ below some threshold will be declared local.) In combination with the energy eigenvectors, one can then compute the $Q$-matrix (2.26), which is most easily done by summing over the local directions, typically much less numerous than the nonlocal ones. Minimizing the complexity (2.30) amounts to finding the lattice point $k_n$ closest to the vector $\frac{E_n t}{2\pi}$ at every time step, with distances defined by the metric $\mathbf{I} + (\mu - 1)\mathbf{Q}$. This is conveniently recast as a standard CVP (3.1) in the Euclidean norm with a redefined basis. Applying the LLL-reduction algorithm to this new lattice basis provides us with a better input for the Babai nearest plane algorithm, which produces a good approximate solution to the CVP. Finally, one can feed this approximate solution to the greedy algorithm, typically resulting in further improvement of the output. At time moment $t$, this procedure finds a CVP candidate solution, which is a lattice point that can be re-expressed in terms of the original hypercubic basis used in (2.30) as a set of integer values $k_n^*(t)$. With these values, we get an explicit, algorithmically implemented complexity bound

$$\mathcal{C}_{\text{bound}}(t) = \left( \sum_{mn} \left[ E_n t - 2\pi k_n^*(t) \right] \{ \delta_{nm} + (\mu - 1) Q_{nm} \} \left[ E_m t - 2\pi k_m^*(t) \right] \right)^{1/2}. \tag{3.17}$$

This approximation for the upper bound on complexity (2.30) is what shall be used in practice in all of our considerations below.

What can one say, most generally, about the behavior of this bound? First of all, in parallel to the bi-invariant case, at times much smaller than $\sqrt{\mu}$, the optimal solution is simply $k_n^*(t) = 0$, which results in the universal linear growth

$$\mathcal{C}_{\text{bound}}(t) = t \quad \text{when} \quad t \ll \sqrt{\mu}, \tag{3.18}$$

keeping in mind the normalization (2.19-2.20), as well as (2.28). Indeed, assigning a single nonzero $k$-value would generically produce a contribution of order $\sqrt{\mu}$ to $\mathcal{C}$, which evidently exceeds the above estimate $t$.

This initial linear growth must evidently saturate. In fact, a straightforward naive bound on $\mathcal{C}$ exists valid at all times. Since the eigenvalues of the $Q$-matrix are all between 0 and 1, as per (2.27), it is certainly true that

$$\mathcal{C}_{\text{bound}}(t) \le \sqrt{\mu} \left[ \sum_n \left( E_n t - 2\pi k_n \right)^2 \right]^{1/2}, \tag{3.19}$$

for any $k_n$. But, up to the factor $\sqrt{\mu}$, the minimum of the right-hand side over $k_n$ is just the bi-invariant complexity of section 2.2. By the same logic as in our analysis of the bi-invariant complexity, each term in the $n$-sum can be made not bigger than $\pi^2$ by choosing $k_n$ appropriately, and hence

$$\mathcal{C}_{\text{bound}}(t) \le \pi \sqrt{\mu D}, \qquad \text{for all } t. \tag{3.20}$$

The actual saturation value is, of course, below this ceiling.[6] For the case of bi-invariant complexity, the saturation value was essentially universal and given by $1/\sqrt{3}$ times the above

---

[6]We briefly note that the conjugate point analysis of [20] suggested that, for chaotic models, no conjugate point would terminate the initial linear complexity growth earlier than times $t \sim \mu/\Delta_{\text{max}} = D/\Delta_{\text{max}}$, with $\Delta_{\text{max}}$ the difference between the largest and smallest energy eigenvalue. The associated complexity plateau can be roughly estimated by writing $\mathcal{C}(t) = \sqrt{\sum_n (E_n t)^2}$ for the initial linear growth. Assuming that individual energy eigenvalues scale in a same way with $\mu = D$ yields the approximation $\mathcal{C}^{sat} \sim \sqrt{\sum_n \mu^2} = \mu\sqrt{D} = D^{3/2}$. Comparing this saturation value with the upper bound (3.20) seems to suggest that, for chaotic dynamics, the termination of the initial linear growth occurs before conjugate points come into play and must therefore be caused, in the parlance of [20, 21], by geodesic loops.

bound (with $\mu = 1$). The main result of this paper is that the behavior of the algorithmically implemented bound (3.17) is more refined, and the complexity plateau height is actually sensitive to the system one studies.

We mention that, while all of our efforts have concentrated on constraining Nielsen's complexity from above, it could also be beneficial to develop lower bounds, which we shall not pursue here. Lower bounds on Nielsen's complexity have been discussed in [16]. It could also be of interest to have useful lower bounds on the solution of the minimization problem (2.30). While such lower bounds will no longer be in a controllable mathematical relation to the original definition of Nielsen's complexity (since they are lower bounds on an upper bound), they could shed further light on the ability of the techniques we develop here to extract physical information of interest. Lower bounds on the exact solution of CVP have been considered, independently of our current perspective, in [96] using the method of Lagrangian relaxation.

## 3.4 Maximal and typical distance from a lattice

For the case of bi-invariant complexity, we arrived at a very neat picture of the complexity plateau by assuming that the point $E_n t$ at a typical (late) moment of time behaves in terms of its distance to the nearest lattice point as a typical point (random, uniformly distributed) in $\mathbb{R}^D$. It could be good to develop the corresponding picture for the complexity bound (2.30), which carries more physical information than the bi-invariant complexity, as we shall see in our numerical studies.

For the bi-invariant complexity (2.17), the relevant CVP was for a hypercubic lattice in the Euclidean norm, and in that case, one can easily prove that the typical distance concentrates at large $D$ on $1/\sqrt{3}$ times the maximal possible distance to the nearest lattice point. While one generally expects similar concentration phenomena for other lattices, we are not aware of a full treatment. Some results that are of relevance for us can be found in [97]. We summarize below some basics in relation to the maximal and typical distances from general lattices.

The maximal distance from a point in $\mathbb{R}^D$ to the nearest lattice point is known in the mathematical literature as the *covering radius*. This name reflects the geometric picture that it is the minimal radius of a sphere such that if one such sphere is centered at every lattice point, all points in space are covered. We can formally define the covering radius $\rho$ as

$$\rho(\mathcal{L}) = \max_x d(x, \mathcal{L}), \tag{3.21}$$

where the maximum is taken over all $x \in \mathbb{R}^D$ and $d(x, \mathcal{L})$ denotes the distance of $x$ to the closest lattice point. As to the average distance from a point uniformly distributed in $\mathbb{R}^D$ to the nearest lattice point, it was proved in [97] that this average distance is always not less than $\rho/2$ (while it is evidently less than $\rho$), and it was also conjectured that it must be not less than $\rho/\sqrt{3}$ (for Euclidean distances that are of interest for us here). We shall use $\rho/\sqrt{3}$ as an estimate for the average distance (and compare it with the observed complexity plateau height with good results), though this is heuristic since we cannot prove concentration around this value mathematically.

Another question is how to estimate the covering radius. Finding it exactly appears a difficult computational problem when $D$ is large (as are many other lattice-related problems); some discussions of its computational complexity can be found in [97]. It turns out, however, that a simple upper bound on the covering radius exists [91] in terms of the Gram-Schmidt orthogonalized basis vectors (3.3). The bound is essentially a corollary of the Babai nearest plane algorithm described in section 3.2. Indeed, the Babai algorithm descends through hyperplanes of lower and lower dimensions by recursively applying the shifts $(l_i - \lfloor l_i \rfloor)b_i^*$ as in (3.12). Since $|l_i - \lfloor l_i \rfloor| \leq 1/2$ and all of the vectors $b_i^*$ are orthogonal to each other (so that the square of the distance from the starting point to the final lattice point reached is the sum-of-squares of the displacements at each step), we get the following upper bound on the distance

from any point to the lattice point Babai algorithm finds, and hence on $\rho$:

$$\rho(\mathcal{L}) \leq \frac{1}{2} \Big( \sum_i ||b_i^*||^2 \Big)^{1/2}. \tag{3.22}$$

The reference value on the right hand side is easy to compute given a lattice basis as defined under (3.1). Then, taking $1/\sqrt{3}$ times this upper bound for the covering radius, and multiplying it by the factor of $2\pi$ on the right-hand side of (3.1) produces a crude estimate for the complexity plateau height:

$$\mathcal{C}_{\text{average estimate}} = \frac{\pi}{\sqrt{3}} \Big( \sum_i ||b_i^*||^2 \Big)^{1/2}. \tag{3.23}$$

We shall see below that this estimate stands comparisons with the numerics rather well. Further improvements in the mathematical understanding of the concentration of distance from general high-dimensional lattices may also put this heuristic estimate on a more solid theoretical footing.

# 4  Complexity bounds for polynomial fermionic Hamiltonians

To test our ideas, it is natural to apply them first to cases that have been studied before, and see how the outcome compares to other methods. In this regard, an adequate starting point is provided by the polynomial fermionic models studied from the complexity viewpoint in [20, 21]. While the main focus of [20, 21] is on identifying conjugate points along the geodesics rather than on direct complexity estimates, some considerations were made in [21] that are rather similar in spirit to our upper bounds, though in a manner that crucially relies on some unique features of the systems treated there. In this section, we shall describe how this dedicated analysis of [21] and the complexity bounds that come from it compare to a blind application of our generic methods to the same systems. We shall see that our methods perform very well and, in some cases, substantially improve the upper bounds proposed in [21].

## 4.1  Integrable and chaotic fermionic models

The studies of [21] were applied to three different types of polynomial fermionic Hamiltonians known as SYK models [38, 39]: 1) free (bilinear) SYK Hamiltonians, 2) quartic integrable deformations thereof and 3) quartic chaotic SYK with generic (random) mode couplings. We shall start by reviewing the mathematical details of these models. The main building blocks are finite dimensional representations of the Clifford algebra

$$\{\psi_i, \psi_j\} = \delta_{ij}, \tag{4.1}$$

for $i, j = 1, \ldots N$. The construction of these representations as $2^{N/2} \times 2^{N/2}$-matrices is straightforward and detailed in e.g. [98].

A basis of operators (observables) in these models can be constructed by considering arbitrary products of $\psi$'s. Since, due to the fermionic nature of $\psi_i$, each index $i$ can appear either once or not at all, this gives $2^N$ possible linearly independent operators. We shall denote as $T_a$ the basis of such operator monomials made of products of $\psi$'s, and normalize it as

$$\text{Tr}[T_a T_b] = \delta_{ab}. \tag{4.2}$$

The definition of locality adopted in [21] considers $T_a$ local if the number of $\psi$'s in the corresponding $\psi$-monomial does not exceed that in the Hamiltonian (we shall be considering

quadratic and quartic Hamiltonians). This 'penalty plan' is sufficient for the models of this section, but we shall consider more general nonlocality assignments further on in our treatment. For the sake of comparison with [21], we will restrict the operators and geodesics to remain within $SU(2^{N/2})$, i.e., in the manifold of unitaries whose determinant is 1. In particular, this means that the identity operator will not be among the local operators when defining the $Q$-matrix of (2.26), and the minimization problem over the family of curves (2.16) is restricted to $\sum_n k_n = 0$. In practice, one can constrain the minimization (2.30) to run over this restricted set of curves by adding a very large penalty to the direction defined by the all-one vector in the space of $k_n$. Since the all-one matrix $\mathbf{1}$ has a unique nonzero eigenvalue in precisely this direction, one can simply modify the metric as $\mathbf{I} + (\mu - 1)\mathbf{Q} \to \mathbf{I} + (\mu - 1)\mathbf{Q} + \nu\mathbf{1}$ with $\nu$ a large number compared to $\mu$, which effectively enforces the tracelessness condition $\sum_n k_n = 0$.

We now state the specific structure of three classes of SYK Hamiltonians acting on Hilbert spaces with dimension $D = 2^{N/2}$:

- Free SYK models are associated to a quadratic (2-local) Hamiltonian

$$H_0 = i \sum_{i,j=0}^{N} J_{ij} \psi^i \psi^j \,, \tag{4.3}$$

with $J_{ij}$ an arbitrary antisymmetric matrix (ensuring the hermiticity of the Hamiltonian). To display some of the conservation laws and spectrum of the free Hamiltonian in a simple manner, one can rotate the matrix $\mathbf{J}$ to a new basis $\mathbf{J} = \mathbf{V}\mathbf{D}\mathbf{V}^T$, where

$$D = \begin{pmatrix} 0 & \omega_1 & 0 & 0 & \cdots \\ -\omega_1 & 0 & 0 & 0 & \cdots \\ 0 & 0 & 0 & \omega_2 & \cdots \\ 0 & 0 & -\omega_2 & 0 & \cdots \\ \vdots & \vdots & \vdots & \vdots & \ddots \end{pmatrix}, \tag{4.4}$$

and $\mathbf{V}$ is an orthogonal matrix (for details regarding the construction of $\mathbf{V}$, we refer to [21]). From this decomposition, it follows that (4.3) can be written as

$$H_0 = 2i \sum_{p=1}^{N/2} \sum_{i,j=1}^{N} V_{i,2p-1} \omega_p V_{j,2p} \psi^i \psi^j \,. \tag{4.5}$$

Defining

$$\Psi_i = \sum_{j=1}^{N} \psi^j V_{ji} \,, \tag{4.6}$$

which satisfy

$$\{\Psi_i, \Psi_j\} = \delta_{ij} \,, \tag{4.7}$$

and

$$A_p = \frac{1}{\sqrt{2}} \left( \Psi_{2p-1} + i\Psi_{2p} \right) \,, \qquad A_p^\dagger = \frac{1}{\sqrt{2}} \left( \Psi_{2p-1} - i\Psi_{2p} \right) \,,$$

$$J_0^{(p)} = I \,, \qquad J_+^{(p)} = A_p^\dagger \,, \qquad J_-^{(p)} = A_p \,, \qquad J_3^{(p)} = A_p^\dagger A_p - A_p A_p^\dagger \,, \tag{4.8}$$

the Hamiltonian becomes

$$H_0 = \sum_{p=1}^{N/2} \omega_p J_3^{(p)} \,. \tag{4.9}$$

In [21], the operators $\left\{J_0^{(p)}, J_-^{(p)}, J_+^{(p)}, J_3^{(p)}\right\}$ are used as building blocks to construct a basis of operators

$$J_{\beta_1}^{(1)} J_{\beta_2}^{(2)} \cdots J_{\beta_{N/2}}^{(N/2)}, \tag{4.10}$$

with $\beta_i = \{0, -, +, 3\}$. One considers $J_3^{(p)}$ as 2-local and $J_\pm^{(p)}$ as 1-local (and discards the identity operator, obtained by setting $\beta_i = 0$ for all $i$'s, as we are restricting the evolution to unit determinant unitaries). We note that this definition agrees with the prescription for constructing a basis of operators classified according to their locality we described below (4.2), since the two formulations are simply related by linear transformations.

Note that the $J_3^{(p)}$ are mutually commuting, while $\{A_p, A_{p'}^\dagger\} = \delta_{pp'}$, so that the operators $J_3^{(p)}$ have eigenvalues -1 and 1 and share the Hamiltonian eigenbasis. The energy spectrum of the free SYK Hamiltonian (4.3) therefore simply consists of the $2^{N/2}$ combinations

$$\{E_n\} = \left\{\pm\omega_1 \pm \omega_2 \pm \cdots \pm \omega_{N/2}\right\}. \tag{4.11}$$

- Integrable (quartic) deformations of the free Hamiltonian are obtained by the addition of a quadratic form in the conserved operators $J_3^{(p)}$:

$$H_{in} = \sum_{p=1}^{N/2} \omega_p J_3^{(p)} + \epsilon \sum_{1 \le p < p' \le N/2} M_{pp'} J_3^{(p)} J_3^{(p')}. \tag{4.12}$$

Here, $M_{pp'}$ is an arbitrary strictly lower triangular matrix, with entries drawn from a Gaussian distribution with mean zero and variance $\sigma^2 = 3! J^2 / N^3$, and $\epsilon$ is the deformation parameter.

- In addition to these two integrable SYK models, we will also consider the following 'chaotic' (generic) deformations [99] of the free model [38, 39]

$$H_{4\text{-body}} = H_0 + \epsilon \sum_{1 \le i < j < k < l \le N} J_{ijkl} \psi^i \psi^j \psi^k \psi^l, \tag{4.13}$$

$$H_{3\text{-body}} = H_0 + i\epsilon \sum_{1 \le i < j < k \le N} J_{ijk} \psi^i \psi^j \psi^k, \tag{4.14}$$

where the coefficients $J_{ijkl}$ and $J_{ijk}$ are drawn from a Gaussian distribution with mean zero and variance $\sigma^2 = 3! J^2 / N^3$ and $\sigma^2 = 2! J^2 / N^2$, respectively. In our numerics, we set $J = 1$.

The complexity bounds developed in [21] start, as does our analysis, with considering the family of curves (2.16), but the handling of the resulting optimization problem is different from ours, and is specific to the free and integrable SYK models. In [21], only those values of the integer parameters $k_n$ are permitted that result in a purely local operator $H'$. Such choices of $k_n$ do not exist generically, but they are present in the free and integrable SYK models defined above, as explained in [21] and reviewed below. Our complexity bound (2.30), on the other hand, does not require any prior knowledge of the analytic properties of the Hamiltonian, and relies on standard lattice optimization algorithms, applied in a universal manner to all systems one studies. In the remainder of this section, we review the local conservation laws of the free and integrable SYK models and compare the performance of our bound (2.30) to the results reported in [21].

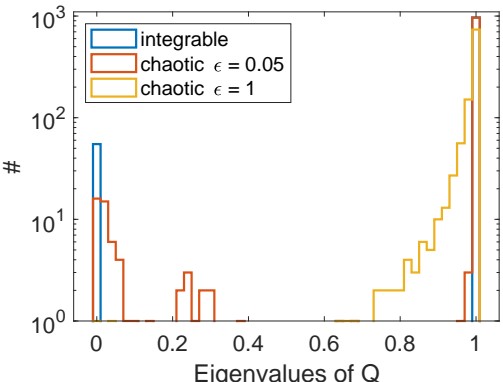
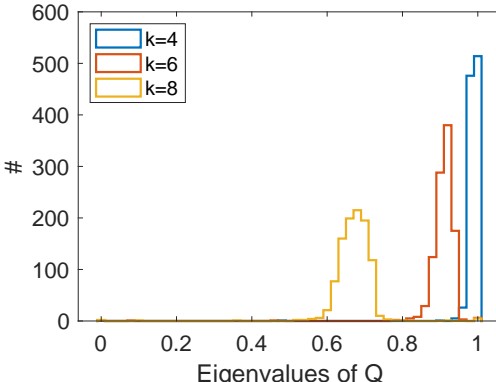

Figure 2: Histograms of the $Q$-eigenvalues for $N = 20$ for the integrable (the free (4.3) and interacting models (4.12) display the same histograms) and quartic chaotic deformations (4.13). **Left:** Histograms for an integrable deformation with $\epsilon = 1$ and two chaotic deformations with $\epsilon = 0.05$ and $\epsilon = 1$ at $k = 4$. **Right:** Histograms for chaotic deformations with $\epsilon = 5$ for different values of the locality threshold $k$.

## 4.2 Conservation laws and complexity reduction

Before discussing the complexity curves obtained using the lattice optimization methods we have described, we would like to compare the features of the $Q$-matrix for the integrable models (4.3) and (4.12) with the chaotic Hamiltonian (4.13). In Fig. 2, we display the distribution of eigenvalues of the $Q$-matrix defined in (2.26) for the integrable and chaotic SYK models. The two types of integrable models (free and quartic) exhibit exactly the same features: all $Q$-eigenvalues are either zero or one. This distribution is highly non-generic and is related to the very large and special set of conservation laws for the integrable SYK models.

One can understand the zero eigenvalues of the $Q$-matrix and their relation to local conservation laws in more detail. As per (2.28), the $Q$-matrix of any model (chaotic or integrable) has at least one null direction defined by the vector of energies. Similarly, if $v_i$ is a null direction of $Q$ ($Qv = 0$), the operator

$$V = \sum_{n=1}^{D} v_n |n\rangle\langle n|, \tag{4.15}$$

is purely local (has zero projections on all nonlocal generators), while it is also evidently conserved, since its eigenvectors $|n\rangle$ by construction coincide with those of the Hamiltonian, and hence $[H, V] = 0$.

That both free and integrable models, for the same locality threshold, share the same number of null directions for $Q$ (and of eigenvalues equal to 1) follows from the fact that their Hamiltonians are built exclusively from the operators $J_3^{(p)}$. Remember that the operators $J_3^{(p)}$, with $p = 1, \cdots, N/2$, are diagonal in the Hamiltonian eigenbasis and square to the identity. Moreover, considering all the inequivalent products similar to (4.10)

$$J_{\beta_1}^{(1)} J_{\beta_2}^{(2)} \cdots J_{\beta_{N/2}}^{(N/2)}, \tag{4.16}$$

restricted to $\beta_i = \{0, 3\}$, this provides a complete and orthogonal basis for the operators that commute with the Hamiltonian. Notice that this structure is very special, since all of the operators (4.16) are homogeneous polynomials in $\psi_i$, as can be seen from (4.6) and (4.8), and whatever locality threshold is chosen, each operator is either purely local or purely nonlocal. The purely local conservation laws then correspond to the null directions of $Q$, while all the remaining ones correspond to the $Q$-eigenvalues 1. Needless to say, this structure is rather

exceptional, as one in general expects that the operators that commute with the Hamiltonian mix contributions of different locality degrees.

The distribution of $Q$-eigenvalues for the chaotic models in Fig. 2 displays more extended structures. For a small deformation parameter $\epsilon$, the null eigenvalues of the free model are smeared towards slightly higher values, since the conservation laws of the free model are no longer exact. As $\epsilon$ increases, however, these small $Q$-eigenvalues do not persist, and when the deformation in (4.13) becomes dominant, one observes that the distribution of $Q$-eigenvalues is sharply concentrated near 1, and the only zero $Q$-eigenvalue remaining is associated to the vector of energies.

It can be observed on the right plot of Fig. 2 that the peak of eigenvalues near 1 moves towards lower values as $k$ increases. This behavior is expected in general: as the locality threshold increases, fewer and fewer generators are treated as nonlocal and get a chance to contribute to the $Q$-matrix. Heuristically, designating a larger number of directions as easy (hence less costly) reduces the height of the complexity plateau, which is also related to the number of small $Q$-eigenvalues.

We now turn to the complexity curves of the three fermionic models, resulting from approximating the solution to the closest vector problem (2.30) using the Babai nearest plane algorithm in the LLL basis, followed by the greedy algorithm. We remind the reader that throughout the paper we rescale the Hamiltonians so that they have norm 1. This choice fixes the initial linear growth of the complexity to have a unit slope and facilitates the comparison between different models and values of $N$.

In Fig. 3, we show the upper bounds on complexity we obtain as a function of time for the free (with locality threshold $k = 2$), interacting but integrable ($k = 4$) and chaotic quartic ($k = 4$) SYK Hamiltonian. On the first two plots, we compare[7] our bounds with corresponding results for the free and interacting integrable model derived in [21]. In that paper, a very stringent upper bound was derived (believed to be the exact complexity value) for the free SYK model (4.3), and a more general upper bound proposed for the interacting integrable model. The observed scaling was $\mathcal{C}(t) \sim \sqrt{D}\left(\log_2 D\right)^{1/2} \sim \sqrt{N} 2^{N/4}$ for the free SYK model, while the upper bound for the complexity of the integrable models scaled as $\mathcal{C}(t) \sim \sqrt{D} \log_2 D \sim N 2^{N/4}$. In comparison with the expectation that chaotic models have a complexity plateau height of order $D$, which is supported by the bottom right plot in Fig. 3, these integrable models therefore exhibit significantly lowered complexity. We remark that these specific scalings hinge on the very peculiar properties of the integrable SYK models, and their broad implications for more generic integrable systems remain open for speculation. (We shall present below studies of complexity bounds for integrable systems with more conventional properties, but these upper bounds also leave much room for possible behaviors of Nielsen's complexity.) In addition, Fig. 3 displays a comparison of different combinations of lattice optimization techniques in application to the integrable interacting SYK model, where one observes that the LLL algorithm dominates the complexity reduction in this specific case.

To compare the two approaches, it is useful to review the ideas used in [21] to establish the bound, and to recast the derivations in the language of the family of curves (2.16) to display

---

[7]Note that in our conventions the complexity is given by $t\|H'\|$ with $H'$ defined as in (2.15) with the vector $k_n$ obtained from the different approximation schemes applied to achieve the minimization (2.30). This definition of complexity differs by a normalization factor $\sqrt{D} = 2^{N/4}$ from the convention adopted in [20,21], as can be seen in, e.g., (4.3) of the former reference. The conventions of [20,21] were chosen to match those of [12], which in turn were designed to relate Nielsen's complexity to a lower bound on the minimal number of quantum gates needed to implement a given $n$-qubit unitary operator. One can easily translate our results in Nielsen's convention using $\mathcal{C}_{\text{Nielsen}}(t) = \mathcal{C}(t)/\sqrt{D}$. In that convention, it is natural to modify the normalization of the Hamiltonian (2.19) to $\text{Tr}[H^2] = D$, which ensures a universal early-time growth $\mathcal{C}_{\text{Nielsen}}(t) = t$ and effectively rescales the $x$- and $y$-axes identically in all our figures displaying the evolution of complexity (since this normalization implies a simultaneous rescaling of the time coordinate).

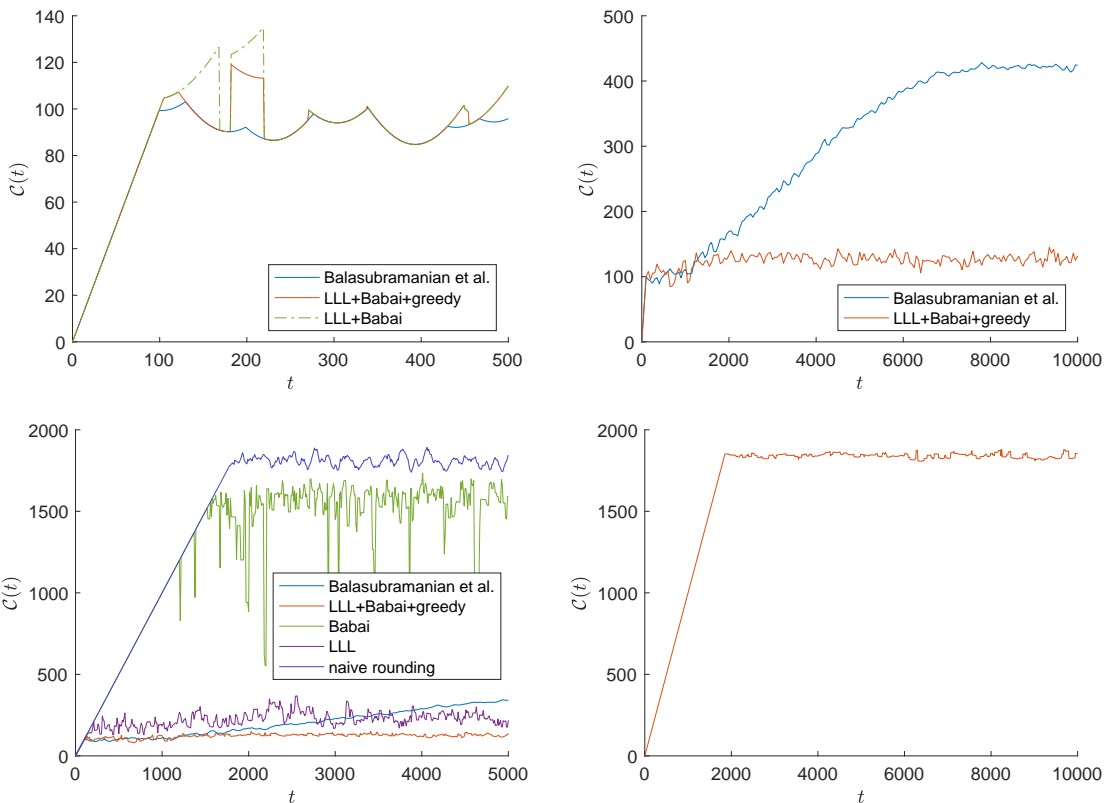

Figure 3: At the top, we display a comparison between the results of [21] (in blue) and the upper bound on complexity (3.17) proposed in the present paper (in red), for **(left)** a realization of the free SYK Hamiltonian with $k = 2$ (4.3). **(right)** a realization of the integrable SYK Hamiltonian (4.12) with $k = 4$ and $\epsilon = 1$. For the free case, the techniques involved in the upper bound (3.17) automatically produce a complexity curve that essentially matches the one derived in [21] which was obtained using model-specific arguments. It can be observed that the greedy descent method improves the bound though the strong suppression in the complexity is already present using only the LLL reduction algorithm and the nearest plane method. For the integrable interacting model, our method achieves a considerably better upper bound than the upper bound proposed in [21]. At the bottom on the left, we display a comparison between different approximation methods for minimizing (2.30) for the integrable interacting SYK model at $k = 4$ and $\epsilon = 1$. The curves listed from top to bottom at late times are given by naive rounding in the standard hypercubic lattice basis (3.9), the Babai algorithm without using the LLL-reduced basis, the upper bound derived in [21], the rounding procedure (3.9) in the LLL-reduced basis and finally, the combination of the LLL algorithm, the nearest plane method and the greedy algorithm. The bottom right figure displays the results of our upper bound for a chaotic realization of (4.13) with locality threshold $k = 4$ and $\epsilon = 5$, where the plateau is considerably higher than in the integrable cases. The behavior displayed in this last plot is expected to be generic, in contrast to the highly peculiar situation characterized by the integrable curves. In all the plots the number of fermion modes is $N = 20$ ($D = 2^{10}$).

their relation to the complexity bound (2.30).

In the free model (4.3), the locality threshold is put to $k = 2$ since the Hamiltonian is quadratic in $\psi_i$. Following the discussion around (4.16), there are $N/2$ local operators $J_3^{(p)}$ that define null directions of the $Q$-matrix. In particular, since the operators $J_3^{(p)}$ have integer eigenvalues equal to 1 and $-1$, this means that one can use these null directions to reduce the first term in (2.25) significantly, without increasing the second term that comes with a large factor $\mu$. Put differently, in free SYK the complexity departs from the line $C(t) = t$ earlier than for generic systems owing to the identity

$$e^{-iH_0 t + 2\pi i \sum_p n_p J_3^{(p)}} = e^{-iH_0 t}, \qquad (4.17)$$

with $n_p \in \mathbb{Z}$ and $H_0$ given by (4.3). Using the expression (4.9) for the Hamiltonian, it is straightforward to see that the addition (4.17) amounts to sending $\omega_p t \to \omega_p t + 2\pi n_p$. Moreover, using the relation (4.11) between the $\omega_p$'s and the energy eigenvalues, the operators $H_0 - 2\pi \sum_p n_p J_3^{(p)}/t$ precisely have the form of velocities (2.15), purely local in this case, with specific combinations of integers $k_n$ that follow from the $n_p$-shifts. Since the additions (4.17) define purely local translations in the space of unitaries, the lengths are simply given by

$$\mathcal{C}(t) = \min_{k_n} \left( \sum_{n=0}^{D} (E_n t - 2\pi k_n)^2 \right)^{1/2}, \qquad (4.18)$$

which for $k_n$ resulting from the shifts (4.17) might be smaller than for $k_n = 0$. Therefore, to minimize (4.18), restricted to shifts of the type (4.17), one should apply additions of the type (4.17) until all $\omega_p t + 2\pi n_p$ are in the interval $]-\pi, \pi]$.

We then define

$$c_p(t) = \omega_p t \mod 2\pi, \qquad (4.19)$$

where the result of the modulo operation is understood to lie in the interval $]-\pi, \pi]$, and the shifted Hamiltonian

$$H' = \sum_p c_p(t) J_3^{(p)}, \qquad (4.20)$$

in the spirit of (2.15), to minimize (4.18) at every time step.

It was shown in [21] that one can in fact reduce the free model complexity even further. It may seem that the rounding procedure (4.19), where one subtracts from the vector $E_n t/2\pi$ the nearest lattice point generated by the eigenvalues of the operators $J_3^{(p)}$, is optimal in view of the orthogonality of $J_3^{(p)}$. However, rational combinations of these operators may have integer eigenvalues and smaller operator norms than $J_3^{(p)}$ themselves. This would simply mean that the operators $J_3^{(p)}$, while being orthogonal, do not generate the entire 'local' sublattice in the corresponding full $k$-lattice (2.15), but only a subset of this local sublattice. Simple examples of such rational combinations are

$$\frac{J_3^{(p_1)} + J_3^{(p_2)}}{2} \qquad \text{and} \qquad \frac{J_3^{(p_1)} - J_3^{(p_2)}}{2}, \qquad (4.21)$$

which have eigenvalues 1, 0, and -1. Therefore, writing

$$c_{p_1}(t) J_3^{(p_1)} + c_{p_2}(t) J_3^{(p_2)} = \left( c_{p_1}(t) + c_{p_2}(t) \right) \frac{J_3^{(p_1)} + J_3^{(p_2)}}{2} + \left( c_{p_1}(t) - c_{p_2}(t) \right) \frac{J_3^{(p_1)} - J_3^{(p_2)}}{2}, \qquad (4.22)$$

one can see that in a situation where both $c_{p_1}(t)$ and $c_{p_2}(t)$ are within the interval $]-\pi, \pi]$ while their sum lies outside that interval, there is remaining freedom offered by (4.21) to reduce

the norm of $H'$ by removing $\pi$ from each of the coefficients, while leaving their difference unaltered. Hence, on top of updating the coefficients $c_p(t)$ of $H'$ so that these remain in the interval $]$-$\pi, \pi]$, one should also update pairs of coefficient in such a way that their sum and difference remain in the interval $]$-$\pi, \pi]$. The rules for implementing this optimization process have been spelled out in [21]:

- first, one generates $c_p(t)$ by starting with $\omega_p t$ and adding to it integer multiples of $2\pi$ as necessary to put it within the interval $]$-$\pi, \pi]$.

- sums of pairs of coefficients $c_{p_1}(t) + c_{p_2}(t)$ should be brought into the interval $]$-$\pi, \pi]$ at all times, adding an integer multiple of $\pi$ to both $c$'s if necessary, keeping their difference unchanged.

- differences of pairs of coefficients $c_{p_1}(t) - c_{p_2}(t)$ should be brought into the interval $]$-$\pi, \pi]$ at all times, adding an integer multiple of $\pi$ to $c_{p_1}(t)$ and subtracting the same number from $c_{p_2}(t)$, keeping the sum $c_{p_1}(t) + c_{p_2}(t)$ unchanged.

It was argued in [21] that this procedure results in finding the exact complexity value, rather than merely an upper bound, also based on comparisons with conjugate point considerations. Comparisons with our lattice optimization bound further support this view.

Using the set of rules described above, we reproduce the complexity curve as computed in [21] for the free SYK model, and show that curve next to our upper bound in the left plot of Fig. 3. One observes that our upper bound captures the actual plateau height of the complexity curve very well. Since the two curves overlap in many places, the proposed upper bound in fact provides an efficient and automated method to identify the directions in which the complexity can be reduced significantly. Comparing the different layers of our method, one can see that the bound already tracks the exact result well before the application of the greedy algorithm, while the addition of the greedy descent method closes the gaps here and there.

For the interacting integrable model, an upper bound on the complexity was proposed in [21] based on similar ideas to the free theory. The locality threshold is now set to $k = 4$, in accordance with the polynomial degree of the interacting Hamiltonian. Since the Hamiltonian is still constructed from the operators $J_3^{(p)}$ only, its conservation laws are the same as in the free case. However, with the new locality threshold assignment, there are more local conservation laws, including the products $J_3^{(p)} J_3^{(q)}$. Similarly to $J_3^{(p)}$, these operators exhibit an integer spectrum of eigenvalues consisting of the values 1 and $-1$. One can therefore exploit this enlarged set of local conservation laws to construct $H'$ of the form

$$H' = \sum_p c_p(t) J_3^{(p)} + \sum_{p<q} m_{pq}(t) J_3^{(p)} J_3^{(q)}, \tag{4.23}$$

with $c_p$ defined as in the free case, and the quartic coefficients $m_{pq}$ constrained to lie within the interval $]$-$\pi, \pi]$:

$$m_{pq}(t) = \epsilon M_{pq} t \mod 2\pi. \tag{4.24}$$

The complexity bound of [21] is then $t||H'||$.

On the right plot of Fig. 3, we compare the resulting upper bound derived in [21] and the upper bound on complexity we propose in this paper, applied to the integrable model (4.12). From this comparison it follows that, like in the free model, our algorithm efficiently deals with the conservation laws of the system and, in fact, lowers the complexity estimate considerably further compared the upper bound developed in [21]. It is natural to ask which additional steps could have been taken in the method described in [21] to achieve the same reduction of the complexity estimate as the automated lower bound that uses the Babai and LLL algorithm,

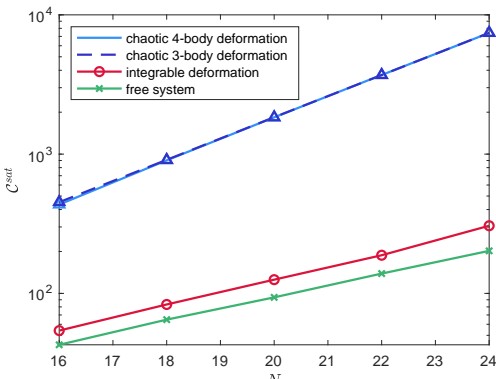 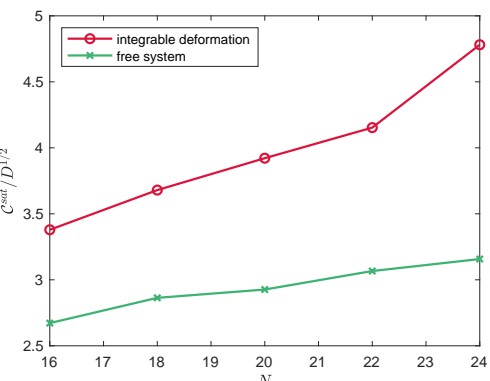

Figure 4: Plots of the saturation value of complexity (left) and the saturation value of complexity divided by $\sqrt{D}$ (right) against $N$ for different systems, locality degree $k = 4$ and for the range $N = 16, 18, \ldots, 24$. On the left, we display, in addition to the integrable and free systems, 3 realizations of each chaotic Hamiltonian using triangles and the spread amongst the chaotic runs can be observed to be very small. The blue lines connect their mean value at every $D$. On the right, we illustrate the slower growth of the integrable models by dividing the complexity curve by $\sqrt{D}$, which removes the exponential dependence on $N$. This normalization matches the complexity convention used in [20, 21], as explained in footnote 7. For all deformations, we set $\epsilon = 1$.

since both methods rely on the intuition that conservation laws (and especially the purely local ones) should contribute to reducing the complexity.

The resolution is that further rational combinations of the local conservation laws with integer eigenvalues can be found, similar to the sums and differences of $J_3^{(p)}$ discussed in the free case. A first reduction can be obtained by considering sums and differences of any pair of local conservation laws, as in (4.22) and not restricted to the 2-local operators. This reduces the complexity significantly, but still leaves it at values higher than the upper bound automatically derived using our methods. We believe the remaining discrepancy can be understood in a similar fashion. Although sums of four local operators in the free theory were discarded in [21] for the reduction of the complexity in the free model, we observed numerically that conserved operators of the form

$$\frac{J_3^p \pm J_3^{p+1} \pm J_3^{p+2} J_3^p \pm J_3^{p+2} J_3^{p+1}}{4}, \tag{4.25}$$

where the number of positive and negative signs must be even, have eigenvalues equal to 1, 0 or -1. This observation shows that further options exist to reduce the upper bound on complexity using an analytic hands-on procedure, bringing it down towards the bound automatically produced by our methods.

To conclude our analysis of complexity in SYK models, we display the dependence of the complexity plateau height on the dimension of the Hilbert space $D$ for the different systems in Fig. 4. For this, we consider a late time window (between $t = 20000$ and $t = 24000$ in steps of $\Delta t = 100$) and compute the mean value of the corresponding complexity values for the four types of SYK models we introduced: the two chaotic and two integrable models. We observe a significant separation between the integrable and chaotic curves. While the chaotic models follow the general expectation that the height of their complexity plateau depends linearly on the Hilbert space dimension $D$, the integrable cases show a strong suppression in complexity that depends, as we have explained, on the very special, non-generic structure of

the conservation laws in these models.

# 5 Quantum resonant systems

In the previous section, we applied our lattice optimization techniques to obtain complexity bounds for a class of fermionic Hamiltonians, and observed that this methodology compares very favorably to the treatments in the past literature. There are strong reasons, however, to pursue the same program for other Hamiltonians, beyond the fermionic SYK models. For one thing, the integrable fermionic models used for the complexity analysis in [21] and the previous section by no means resemble generic integrable systems. They possess very large sets of conservation laws that are quadratic and quartic in the dynamical variables (the number of these conservation laws grows logarithmically without bound as the size of the Hilbert space increases). This is clearly atypical and, as we saw, it is precisely this profusion of polynomial conservation laws that leads to a dramatic reduction of complexity estimates. It could certainly be advantageous to test our techniques in application to integrable systems that could be viewed as more generic. Another issue (which may be more or less relevant depending on the concrete applications one has in mind) is that fermionic systems do not have classical limits. Since 'integrability' and 'chaos' are only sharply defined for classical systems, applying these notions to quantum systems without classical counterparts has its limitations.

With these considerations in mind, we shall now turn to bosonic models to explore the ideas of complexity. One is used to the concept that bosonic models inhabit infinite-dimensional Hilbert spaces, and are largely intractable in the presence of interactions. This obstacle is avoided in the specific class of quartic bosonic models called *quantum resonant systems* that can be seen as a bosonic analogue of the SYK models. Quantum resonant systems (and their classical limits) arise as consistent weakly nonlinear approximations to a variety of physical problems featuring weak nonlinearities in strongly resonant domains. A recent review can be found in [80]. While their Hilbert spaces are infinite-dimensional, the Hamiltonians are block-diagonal in the Fock basis, with all blocks of finite sizes (although there is no bound on the size). Thus, the Hamiltonian can be straightforwardly diagonalized block-by-block, and the eigenvalues and eigenvectors studied explicitly, from the complexity perspective, for example. In contrast to the simplicity of the quantum diagonalization, the corresponding classical dynamics is very rich and has been widely investigated, with attractive integrable examples present for specific choices of the coupling values.

This section is mostly a review of the basic properties of quantum resonant systems and their emergence in physics, following the original exposition of [41] and some other sources. In the next section, we then turn to analyzing the complexity bounds for integrable and nonintegrable representatives of this class of systems.

## 5.1 Resonant Hamiltonians and their block-diagonal structure

Quantum resonant systems are described by the quartic Hamiltonian

$$H = \frac{1}{2} \sum_{\substack{n,m,k,l=0 \\ n+m=k+l}}^{\infty} C_{nmkl} \, a_n^\dagger a_m^\dagger a_k a_l \,, \tag{5.1}$$

with *interaction coefficients* $C_{nmkl}$ that have to satisfy $C_{nmkl} = C_{klnm} = \bar{C}_{nmlk}$ to ensure Hermiticity. The operators $a_n^\dagger$ ($a_n$) with $n \geq 0$ are bosonic creation (annihilation) operators satisfying the standard commutation relations

$$[a_n, a_m^\dagger] = \delta_{nm} \,. \tag{5.2}$$

The Hamiltonian (5.1) acts on an infinite-dimensional Fock space spanned by the vectors

$$|\eta_0, \eta_1, \dots\rangle = \prod_{k=0}^{\infty} \frac{(a_k^\dagger)^{\eta_k}}{\sqrt{\eta_k!}} |0, 0, 0, \dots\rangle, \tag{5.3}$$

where, as usual, $\eta_i$ are the particle occupation numbers in mode $i$.

While generic bosonic quartic interactions would be computationally intractable due to the infinite number of Hilbert space dimensions, the definition (5.1) essentially features the *resonance condition* $n + m = k + l$ that simplifies the Hamiltonian diagonalization procedure dramatically. To see this, we note that the Hamiltonian (5.1) manifestly conserves the particle number[8]

$$N = \sum_{n=0}^{\infty} a_n^\dagger a_n, \tag{5.4}$$

and as a consequence, its matrix elements between states with different particle number vanish. Moreover, the resonance condition $n + m = k + l$ below the summation in (5.1) makes the Hamiltonian commute with the operator

$$M = \sum_{n=0}^{\infty} n\, a_n^\dagger a_n. \tag{5.5}$$

The conserved operators $N$ and $M$ are diagonal in the Fock basis (5.3), with eigenvalues

$$N = \sum_{n=0}^{\infty} \eta_n \qquad \text{and} \qquad M = \sum_{n=0}^{\infty} n\eta_n. \tag{5.6}$$

From this expression, it is clear that, for fixed positive integers $N$ and $M$, there exist only a finite number of possible arrangements of the occupation numbers $\eta_i$ of the vectors (5.3) that satisfy (5.6). The Hamiltonian then only connects the states within such $(N, M)$-block to other states within the same block, making the diagonalization problem effectively finite-dimensional.

It is interesting to contrast the scaling of number of dimensions of the fixed $(N, M)$- subspaces of resonant Hamiltonians with the scaling of fermionic models with $N$ modes. From (5.6), it follows that the size of an $(N, M)$-block is given the number of ways to obtain $M$ by adding $N$ integer values that range from 0 to $M$. This amount equivalently corresponds to the number of partitions of $M$ into at most $N$ parts, a common function in number theory denoted as $p_N(M)$ [100, 101]. Although there is no closed-form expression for this quantity, a few useful asymptotic behaviors are known. For example, when $N = M$, which is the case we shall mostly focus on in applications, $p_N(N)$ simply equals the total number of partitions of $N$, and its large $N$ asymptotics is well-known in number theory as [101]

$$p_N(N) \sim \frac{1}{4N\sqrt{3}} \exp\left(\pi\sqrt{\frac{2N}{3}}\right). \tag{5.7}$$

This latter scaling is much milder than the $2^N$ scaling displayed by fermionic models, which is a welcome feature since one gets a chance to explore a greater variety of the values of $N$ before one hits the computational ceilings.

In practice, one starts with (5.1) and restricts the considerations to a particular chosen $(N, M)$-block. Within that block, the matrix elements of the Hamiltonian will be explicit linear functions of the interaction coefficients $C$. Once the interaction coefficients have been specified, this becomes an explicit numerical matrix of size $p_N(M) \times p_N(M)$. This matrix is straightforwardly diagonalized, producing explicit eigenvalues and eigenvectors, whereupon one can study these eigenvalues and eigenvectors, and/or proceed to another $(N, M)$-block as one chooses.

---

[8]This bosonic particle number $N$ should not be confused with the number of fermion modes $N$ in section 4.

## 5.2 Classical dynamics and integrability

The extremely simple picture of the quantum dynamics we have just spelled out, based entirely on diagonalization of finite-sized numerical matrices, may create an impression that the physics encoded by quantum resonant systems is likewise very simple. This, however, is definitely not so, which is most clearly seen when considering their classical limits.

In the classical limit, the creation-annihilation operators $a_n$ and $a_n^\dagger$ are replaced with complex-valued classical variables $\alpha_n$ and $\bar{\alpha}_n$, so that $\alpha_n$ and $i\bar{\alpha}_n$ are canonically conjugate. The corresponding Hamiltonian equation of motion is

$$i\frac{d\alpha_n}{dt} = \sum_{\substack{k,l,m=0 \\ n+m=k+l}}^{\infty} C_{nmkl}\,\bar{\alpha}_m \alpha_k \alpha_l\,. \tag{5.8}$$

It is known from extensive literature [44–51, 54–77] that, depending on the specific values of $C$, such systems of equations display a striking range of behaviors, some of which are at the forefront of contemporary PDE mathematics. Dynamical features that have been specifically studied are turbulent transfers of energy [71–77] including finite-time turbulent blow-up [44–47, 49, 51, 77], dynamical recurrence phenomena [44, 50, 69], integrability [71–77], extra conservation laws and solvability within restricted dynamically invariant manifolds [54, 57, 58, 64–66, 78, 79], etc. Thus, not only can one reliably connect the extremely simple quantum dynamics of (5.1) with its effectively finite-dimensional Hilbert spaces to classical Hamiltonian dynamics, but in addition this classical dynamics can display an array of rich and varied features depending on the choice of the couplings $C$.

We proceed to present a few special cases of quantum resonant systems that will be of relevance in our studies. An essential aspect for us is to contrast the behavior of random values of $C$, which is generically expected to result in chaotic dynamics typical of nonlinear systems with an infinite number of degrees of freedom, and choices of $C$ that result in integrable dynamics. We start with an extremely special system where all the coefficients $C_{nmkl}$ have been set to 1:

$$H_{\mathrm{GG}} = \frac{1}{2}\sum_{\substack{n,m,k,l=0 \\ n+m=k+l}}^{\infty} a_n^\dagger a_m^\dagger a_k a_l\,. \tag{5.9}$$

This system is the quantized analogue of the classical *cubic Szegő equation*[9] introduced and studied by Gérard and Grellier (hence the abbreviation GG) in [71–74]. The classical dynamics of this system is known to be extremely special, and possesses two independent Lax-pairs, indicating a strong form of integrability that has been used to write down a formal general analytic solution for all classical trajectories. (This solution furthermore displays a range of prominent turbulent behaviors.) The corresponding quantum Hamiltonian (5.9) is in a way even more striking, since it is known numerically [41] that all of its eigenvalues are integer. Thus, we are clearly not dealing with a generic integrable system, but rather with some form of superintegrability, though the precise nature of this superintegrability still requires further elucidation at this point.

In view of the above, while the GG Hamiltonian forms an important point of departure for our studies, it is by itself of limited use, because it is simply too special. In particular, because its spectrum is purely integer, the evolution operator will return to the identity exactly after a fixed short period, which will undermine the existence of a complexity plateau, a key feature of complexity-related phenomenology one expects in generic systems.

---

[9]The name of Szegő appears in relation to the usage of the Szegő projector in the position-space formulation of this model. We shall briefly review it in section 5.5, though this material is by no means essential for our present purposes.

What is important for us is that deformations of the GG Hamiltonian exist [75–77] that break part of the Lax structure of (5.9) but are still classically Lax integrable. (We will review the Lax pair structures in section 5.5, though practically speaking, their concrete form plays no technical role in our current studies, and the most important observation for us here is simply that these Lax pairs exist and signify a specific form of classical integrability.) We shall furthermore see below that their quantum energy spectra display a Poissonian level spacing statistics associated with generic integrable systems [43]. They thus form an advantageous reference point for studying the interplay of integrability and complexity.

The first integrable deformation, called the truncated Szegő model [77], is obtained by removing all interactions in (5.9) that do not involve the zero mode:

$$C_{nmkl}^{(tr)} = \begin{cases} 0 & \text{if } n \neq 0,\ m \neq 0,\ k \neq 0,\ l \neq 0, \\ 1 & \text{otherwise}. \end{cases} \tag{5.10}$$

The corresponding truncated Szegő Hamiltonian can be written as

$$H_{tr} = \frac{1}{2} \sum_{\substack{n,m,k,l=0 \\ n+m=k+l}}^{\infty} a_n^\dagger a_m^\dagger a_k a_l - \frac{1}{2} \sum_{\substack{n,m,k,l=1 \\ n+m=k+l}}^{\infty} a_n^\dagger a_m^\dagger a_k a_l. \tag{5.11}$$

Some of the turbulent properties of the corresponding classical dynamics were studied in [77]. (Curiously, within some classes of initial data, the turbulent behaviors get stronger than in the GG system despite an overwhelming majority of mode coupling having been removed.) This system preserves one of the two Lax pairs of (5.9). There is furthermore a one-parameter family [77] of integrable Hamiltonians that smoothly connect (5.11) and (5.9), hence our usage of the term 'deformation,' but we will not study them here explicitly.

Another integrable deformation of (5.9) is obtained [75, 76] by adding to it a multiple of $a_0^\dagger a_0$:

$$H_\alpha = \frac{1}{2} \sum_{\substack{n,m,k,l=0 \\ n+m=k+l}}^{\infty} a_n^\dagger a_m^\dagger a_k a_l + \alpha\, a_0^\dagger a_0. \tag{5.12}$$

We shall see below that, when $\alpha$ is of order 1, this system still does not behave as a generic integrable system with respect to its spectral statistics, as there are too many small gaps between the energy levels. We will use this situation as a test case representing a 'somewhat special' integrable system. However, if one looks at an $(N, M)$-block with $N = M$ and tunes the value of $\alpha$ up to be close to $M$, the energy levels behave as one would expect for a generic integrable system.

While it may sound awkward in relation to the last situation we have just described to identify $\alpha$ (a parameter in the Hamiltonian) with $M$ (a state-dependent conserved quantity), one can recast this setup in a more conventional language by simply defining a different deformation of (5.9) proposed in [77]:

$$H_\delta = \frac{1}{2} \sum_{\substack{n,m,k,l=0 \\ n+m=k+l}}^{\infty} a_n^\dagger a_m^\dagger a_k a_l + \delta \sum_{n=1}^{\infty} n\, a_n^\dagger a_0^\dagger a_n a_0 = \frac{1}{2} \sum_{\substack{n,m,k,l=0 \\ n+m=k+l}}^{\infty} a_n^\dagger a_m^\dagger a_k a_l + \delta M a_0^\dagger a_0. \tag{5.13}$$

Evidently, within any given $(N, M)$-block, the matrix elements of (5.12) are exactly the same as the matrix elements of (5.13) with $\delta = \alpha/M$.

As usual for integrable systems, all of the above examples come with infinite towers of conservation laws. These laws have been understood very thoroughly using Lax integrability in the classical case for the GG Hamiltonian [71–74], while some less complete understanding

has also been developed for the deformations [75–77]. The quantization of these hierarchies of conservation laws has never been worked out explicitly, but they are expected to form towers made of polynomials of growing degrees in the creation-annihilation operators. To illustrate this point, we quote a rather nontrivial quartic conservation law valid for all the above systems, (5.9), (5.11), (5.12) and (5.13):

$$H_{min} = \sum_{\substack{n,m,k,l=1 \\ n+m=k+l}}^{\infty} \min(n,m,k,l) \, a_n^\dagger a_m^\dagger a_k a_l + \sum_{k=1}^{\infty} k^2 a_k^\dagger a_k \,. \tag{5.14}$$

We have in fact discovered this conservation law experimentally by looking at the null vectors of the $Q$-matrix (2.26). It turns out that the classical counterpart of (5.14), where the quadratic last term disappears upon taking the classical limit, is in a straightforward relation[10] to the Lax pair structure (being simply $\text{Tr}(K_u^4)$, where $K_u$ is on of the Lax operators we shall review in section 5.5). Thus, besides its role in the complexity analysis, the $Q$-matrix provides an algorithmic way to recover polynomial conservation laws in systems with finite-dimensional Hilbert spaces. Interestingly, numerical diagonalization of (5.14) indicates that this operator itself possesses a purely integer spectrum of eigenvalues.

## 5.3 Level spacing statistics

Our subsequent analysis will be dedicated to comparisons of complexity bounds for integrable vs. chaotic systems. As mentioned briefly in section 2.2, one way to distinguish these two types of systems is through their level spacing statistics [42,43]. Conjectures by Berry-Tabor [88] and Bohigas-Giannoni-Schmit [89] control the distribution of energy level spacings for integrable and chaotic cases, respectively. (These conjectures are sometimes used to define what is meant by 'quantum integrability' for systems that do not possess classical limits.)

One important technical ingredient in the construction of level spacing distributions is that the distances between neighboring energies must be properly normalized in a process commonly referred to as 'unfolding'. Unfolding the energy spectrum comes down to locally rescaling the energy so that the mean level density becomes constant on scales much bigger than the individual level spacings. It is such unfolded energy spectra that are expected to possess the universal properties postulated in quantum chaos theory. Many variations of the unfolding process may be designed, but one concrete implementation [41] that is convenient for our purposes is as follows. In order to unfold the spacings of an energy spectrum consisting of $p_M(N)$ energy eigenstates $E_I$ within an $(N, M)$-block of the resonant Hamiltonian, we first compute the raw unfolded spacings $s_I^{(raw)}$:

$$s_I^{(raw)} = \frac{E_{I+1} - E_I}{E_{I+\Delta} - E_{I-\Delta}} \,, \tag{5.15}$$

with $\Delta$ being an integer close to $\sqrt{p_N(M)}$ and $I$ going from $\Delta + 1$ to $p_N(M) - \Delta$. The factor $1/(E_{I+\Delta} - E_{I-\Delta})$ here is proportional to the mean level density of the system (over a range of $2\Delta$), which should become 1 after the unfolding procedure. Next, the unfolded spacings are defined as:

$$s_I = \frac{s_I^{(raw)}}{\bar{s}} \,, \tag{5.16}$$

with $\bar{s}$ being the average of the raw spacings $s_I^{(raw)}$. Note that the mean level density is indeed equal to 1 now since the mean of all $s_I$ equals 1 by construction.

---

[10]We thank Patrick Gérard for clarifying this point to us.



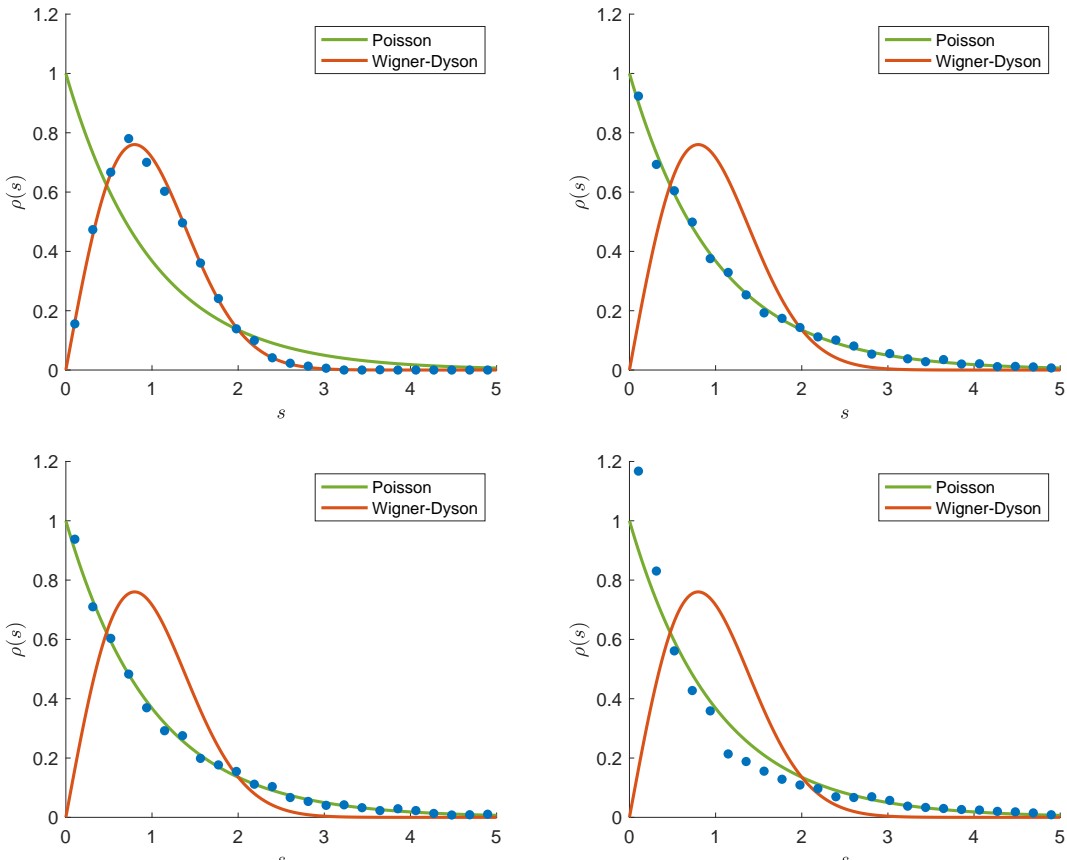

Figure 5: Unfolded level spacing statistics for the block $(N, M) = (30, 30)$ for **(top left)** a realization of a random resonant Hamiltonian; **(top right)** truncated Szegő system; **(bottom left)** $\delta$-deformed Szegő system with $\delta = 1/2$; **(bottom right)** $\alpha$-deformed Szegő system with $\alpha = 1$. The last model displays a more pronounced peak at small level spacings than the Poisson distribution. The likely origin of this feature is the proximity of this system to the original GG Hamiltonian (5.9), which has many exact degeneracies in its purely integer energy spectrum.

The level spacing statistics of chaotic systems is well-approximated by the *Wigner-Dyson distribution*. This distribution originates from random matrix theory, where it is obeyed by the unfolded distances between neighboring eigenvalues in an ensemble of random matrices. The exact Wigner-Dyson distribution does not have a closed form expression in terms of elementary functions, but it is very well-approximated for practical purposes by the following *Wigner surmise*:

$$\rho_{Wigner}(s) = \frac{\pi s}{2} e^{-\pi s^2/4}. \tag{5.17}$$

Note that as $s$ goes to zero, the distribution vanishes, which signifies level repulsion.

To represent chaotic models, we will consider a set of random quantum resonant Hamiltonians defined by interaction coefficients $C_{nmkl}$ drawn from a uniform distribution in the interval $(0, 1)$. We plot the level spacing statistics of one realization of the random quantum resonant model in Fig. 5. It is clear from the figure that the level spacing of our random Hamiltonian indeed follows the Wigner-Dyson distribution [41].

Generic integrable systems obey a different kind of statistics. The energy eigenvalues of such systems are uncorrelated, and the unfolded distances behave like distances between

points randomly thrown on a line, resulting in the Poisson distribution:

$$\rho_{Poisson}(s) = e^{-s}. \tag{5.18}$$

The absence of correlation between the energy levels makes the level repulsion disappear. This is the most prominent distinction between the statistics for integrable and chaotic systems.

The GG Hamiltonian (5.9) does not fit into this scheme as it is too special. Its purely integer energy spectrum contains exact degeneracies that result in a sort of 'super-Poissonian' level spacing distribution that favors degeneracy. The deformations of this Hamiltonian mentioned in section 5.2, on the other hand, work much better in the capacity of generic integrable systems from the standpoint of quantum chaos theory. In addition to the level spacing statistics of a random system (which predictably agrees with the Wigner-Dyson curve), we display in Fig. 5 the corresponding plots for three possible deformations of the GG Hamiltonian. For the truncated Szegő system and $\delta$-deformation with $\delta = 1/2$, we observe very accurate Poissonian spectra. For the $\alpha$-deformation with $\alpha = 1$, the spectrum is 'super-Poissonian' with excess of small level spacings, so this is a somewhat special system. Overall, we find that some deformations of the GG Hamiltonian provide excellent models with features of generic integrable systems, first, due to their accurate Poissonian spectra (often seen as a key feature of quantum integrability), and second, for the explicitly known Lax structures of their classical counterparts (which is the most widely studied form of integrability for systems with infinite numbers of degrees of freedom).

## 5.4   Resonant systems as weakly nonlinear approximations

For the practical purposes of this paper, it is sufficient to treat quantum resonant systems as explicitly defined Hamiltonians and study their properties, in particular, in relation to complexity. It is pedagogically worthwhile, however, to explain how such systems arise from more familiar physical theories, since this may not be obvious at first sight. We shall now proceed with a brief review of this sort, which may be comfortably skipped by practically-minded readers.

Hamiltonians of the form (5.1) naturally emerge in the description of weakly nonlinear dynamics with weak quartic interactions in strongly resonant bounded domains (a review can be found in [80]). We first start with the classical version of the story, and as a very simple particular case consider the nonlinear Schrödinger equation in one dimension with a harmonic potential:

$$i \frac{\partial \Psi}{\partial t} = \frac{1}{2} \left( -\frac{\partial^2}{\partial x^2} + x^2 \right) \Psi + g |\Psi|^2 \Psi. \tag{5.19}$$

A general solution at finite $g$ can be found by expanding the wavefunction in a basis of eigenfunctions of the linearized problem (at $g = 0$)

$$\Psi(x, t) = \sum_{n=0}^{\infty} \alpha_n(t) \psi_n(x) e^{-iE_n t}, \qquad E_n = n + \frac{1}{2}, \qquad \frac{1}{2} \left( -\frac{\partial^2}{\partial x^2} + x^2 \right) \psi_n = E_n \psi_n, \tag{5.20}$$

and allowing the coefficients $\alpha_n$ to depend on time. The time-dependence of these modes is then obtained by substituting (5.20) back into (5.19), and projecting the resulting equation on $\psi_n(x)$. This yields the following flow equation for the modes $\alpha_n$

$$i \frac{d\alpha_n}{dt} = g \sum_{k,l,m=0}^{\infty} C_{nmkl} \, \bar{\alpha}_m \alpha_k \alpha_l \, e^{i(E_n + E_m - E_k - E_l)t}, \tag{5.21}$$

with interaction coefficients $C_{nmkl} = \int \psi_n \psi_m \psi_k \psi_l \, dx$.

One then turns to the weak coupling regime with small $g$. As can be seen from the right hand side of (5.21), when $g$ is small, the nonlinear interaction term induces slow variations in

the evolution of the modes. However, provided one waits long enough, these terms may accumulate to $\mathcal{O}(1)$ effects on long time scales of $\mathcal{O}(1/g)$. The dominant terms in the summation in (5.21) that cause this build-up effect are the terms for which $E_n + E_m - E_k - E_l = 0 = n + m - k - l$. Note that the resonant[11] character of the spectrum of the harmonic oscillator allows for nontrivial cancellations of this type in the exponent. The other terms, for which $E_n + E_m \neq E_k + E_l$, effectively average out on time scales larger than $\mathcal{O}(1)$ and one can therefore consider the following simpler evolution equation for the modes

$$i \frac{d\alpha_n}{dt} = g \sum_{\substack{k,l,m=0 \\ n+m=k+l}}^{\infty} C_{nmkl}\, \bar{\alpha}_m \alpha_k \alpha_l \,. \tag{5.22}$$

These are precisely the classical equations of motion corresponding to the resonant Hamiltonian (5.1). Similar derivations hold for a variety of other physical equations of motion in different numbers of dimensions taken as the starting point [80]. The key ingredients are weak quartic nonlinearities and a spatial domain with a discrete highly resonant spectrum of normal mode frequencies. The underlying physics gets completely encoded in the end as the mode couplings $C$, which are evidently different when applying these methods to different physical systems.

Solutions of (5.22) are known to approximate the solution of the original equation of motion, (5.21) in our present toy example, at leading order in $g$ on long time scales $\mathcal{O}(1/g)$. General pedagogical review of such approximations can be found in [102, 103]. Rigorous mathematical proofs specifically for the one-dimensional example above can be found in [67], and for some other nonlinear Schrödinger equations, in [61].

Turning to the corresponding quantum case, the averaging over 'fast' oscillations applied to the classical equations in (5.21) turns into something more conventional: standard quantum-mechanical perturbation theory in the presence of energy level degeneracy [81–84]. Indeed, free quantum fields in strongly resonant domains possess highly degenerate energy levels, and these degenerate energy levels of free fields are precisely the physical origin of the $(N, M)$-blocks we had discovered by direct inspection of resonant Hamiltonians.

We now demonstrate how this perturbation theory pattern works, again choosing a simple one-dimensional example for the sake of illustration. Consider the quantum Hamiltonian associated with (5.19):

$$\mathcal{H} = H_0 + g H_{int} = \frac{1}{2} \int \left( \frac{\partial \Psi^\dagger}{\partial x} \frac{\partial \Psi}{\partial x} + x^2 \Psi^\dagger \Psi + g\, \Psi^{\dagger 2} \Psi^2 \right) dx \,, \tag{5.23}$$

where $\Psi$ is now a nonrelativistic bosonic field operator. This is just the second-quantized picture of a system of nonrelativistic bosons with two-particle contact interactions in a harmonic potential. We expand $\Psi$ in the basis of eigenfunctions $\psi_n$ given in (5.20)

$$\Psi = \sum_{n=0}^{\infty} a_n \psi_n(x), \tag{5.24}$$

which defines the bosonic annihilation operators $a_n$ satisfying (5.2). We then substitute (5.24) in (5.23) and find for the free Hamiltonian (after subtracting an irrelevant vacuum energy term)

$$H_0 = \sum_{n=0}^{\infty} n\, a_n^\dagger a_n \,, \tag{5.25}$$

---

[11]A spectrum $\{E_i\}$ is called resonant if there exists a set of not simultaneously vanishing integers $n_k$ such that $\sum_k n_k E_k = 0$.

while the interaction Hamiltonian is

$$H_{int} = \frac{1}{2} \sum_{n,m,k,l=0}^{\infty} C_{nmkl}\, a_n^\dagger a_m^\dagger a_k a_l\,. \qquad (5.26)$$

By construction, $H_{int}$ conserves the particle number $N$. Moreover, to find the energy shifts at first order in perturbation theory, one should diagonalize the interaction Hamiltonian (5.26) in degenerate energy eigenspaces of the free Hamiltonian (5.25). Since the eigenstates of $H_0 = M$ at a fixed value of the unperturbed energy are precisely the Fock states (5.3) at fixed value of $M$, the only terms in the sum in (5.26) that have nonvanishing matrix elements between fixed $M$ eigenstates are those constrained to satisfy the resonant condition $n+m = k+l$. For the sake of computing matrix elements between states within the same degenerate energy level of the free system, on can thus replace (5.26) by (5.1) Thereby, the quantum resonant Hamiltonian (5.1) emerges naturally in the process of finding the energy shifts due to adding $gH_{int}$ to $H_0$.

## 5.5 Lax integrability

We add a brief review, for completeness, of the Lax integrability for the GG system and its deformations relevant for our work. This material is an aside, and is not crucial for understanding the practical side of our complexity analysis. What is important for us is that the existence of explicitly known Lax pairs in the classical theory gives these models a privileged status.

We introduced the GG system (5.9) in terms of the modes $a_n$, which become complex-valued variables $\alpha_n$ in the classical version. The Lax pair structure, however, is more manageable in the position space formulation, as in [71–74], where one considers functions $u$ defined on the unit circle in the complex plane.

$$u(t, e^{i\theta}) = \sum_{n=0}^{\infty} \alpha_n(t) e^{in\theta}\,. \qquad (5.27)$$

One furthermore introduces the Szegő projector $\Pi$ that filters out the negative frequency modes:

$$\Pi\Big(\sum_{n\in\mathbb{Z}} \alpha_n e^{in\theta}\Big) = \sum_{n=0}^{\infty} \alpha_n e^{in\theta}\,. \qquad (5.28)$$

Using this projector, the classical equations of motion corresponding to (5.9) become

$$i\frac{\partial u}{\partial t} = \Pi\left(|u|^2 u\right)\,. \qquad (5.29)$$

Writing down the Lax pair explicitly benefits from introducing the following operators:

$$H_u h = \Pi(u\bar{h})\,, \qquad T_b h = \Pi(bh)\,, \qquad Sh = e^{i\theta}h\,, \qquad (5.30)$$

where $S$ is a shift operator that sends a sequence of coefficients $\{\alpha_0, \alpha_1, \ldots\}$ to $\{0, \alpha_0, \alpha_1, \ldots\}$. Similarly, the conjugate operation $S^\dagger$ sends the sequence $\{\alpha_0, \alpha_1, \ldots\}$ into $\{\alpha_1, \alpha_2, \ldots\}$. Note that $H_u$ is an anti-linear operator, rather unconventional for expositions of Lax theory, but it has been used effectively in [71–74], and its even powers become linear operators. In terms of the original modes, these operations can be given as

$$(H_u h)_n = \sum_{m=0}^{\infty} \alpha_{n+m}\bar{h}_m\,, \qquad (T_b h)_n = \sum_{m=0}^{\infty} b_{n-m}h_m\,, \qquad (Sh)_n = h_{n-1}\,. \qquad (5.31)$$

(It is in principle possible to handle the Lax pairs in the mode language, and some such derivations can be seen in Appendix A of [77]. This approach is more cumbersome than the position-space derivations in the style of [71–74], however.)

In [71], the following two Lax identities were derived for $u$ satisfying the cubic Szegő equation (5.29):

$$\frac{dH_u}{dt} = [B_u, H_u], \qquad \frac{dK_u}{dt} = [C_u, K_u], \tag{5.32}$$

with

$$K_u = S^\dagger H_u = H_u S = H_{S^\dagger u}, \qquad B_u = \frac{i}{2} H_u^2 - i T_{|u|^2}, \qquad C_u = \frac{i}{2} K_u^2 - i T_{|u|^2}. \tag{5.33}$$

As usual in the presence of Lax structures, a tower of conserved quantities follows from considering traces of powers of the first Lax operator in each pair. Since $H_u$ and $K_u$ are antilinear operators in our case, only the traces of even powers,

$$\mathrm{Tr}[H_u^{2n}] \qquad \text{and} \qquad \mathrm{Tr}[K_u^{2n}], \tag{5.34}$$

provide conservation laws. In particular, setting $n = 2$ and using the mode representations (5.31) one finds

$$\mathrm{Tr}[H_u^4] = \sum_{\substack{n,m,k,l=0 \\ n+m=k+l}}^{\infty} [1 + \min(n,m,k,l)] \, \bar{\alpha}_n \bar{\alpha}_m \alpha_k \alpha_l, \tag{5.35}$$

$$\mathrm{Tr}[K_u^4] = \sum_{\substack{n,m,k,l=0 \\ n+m=k+l}}^{\infty} \min(n,m,k,l) \, \bar{\alpha}_n \bar{\alpha}_m \alpha_k \alpha_l. \tag{5.36}$$

The second line is precisely the classical version of the quantum conservation law (5.14) we discovered empirically. Note that at every $n$, the associated conservation law is a polynomial of order $2n$ in the modes $\alpha_n$.

We mention that the peculiar structure (5.32) of the GG system results in additional simple conservation laws. This is most easily seen from observing that the squares of $H_u$ and $K_u$ also satisfy Lax equations, but now with the same Lax partner:

$$\frac{dH_u^2}{dt} = -i[T_{|u|^2}, H_u^2], \qquad \frac{dK_u^2}{dt} = -i[T_{|u|^2}, K_u^2]. \tag{5.37}$$

As a consequence, their difference

$$P_u = H_u^2 - K_u^2 = (u|\cdot)u, \tag{5.38}$$

likewise forms a Lax pair with $-i T_{|u|^2}$. The following scalar product has been used in the last expression:

$$(u|v) \equiv \int_0^{2\pi} \bar{u} v \frac{d\theta}{2\pi}. \tag{5.39}$$

Hence, the trace of any product of these three operators defines a conservation law, in particular,

$$\mathrm{Tr}[H_u^{2n} P_u] = (u|H_u^{2n} u), \qquad \text{and} \qquad \mathrm{Tr}[K_u^{2n} P_u] = (u|K_u^{2n} u). \tag{5.40}$$

Explicit quantization of all of these classical conservation laws remains an outstanding problem.

As we move away from the GG Hamiltonian and introduce the truncated Szegő system, as well as the $\alpha$ and $\delta$ deformations of section 5.2, the $H_u$-based Lax pair is destroyed. However, all of these three systems retain the second $K_u$-based Lax pair, while the Lax partner of $K_u$ gets modified to $C_u - B_{S^\dagger u}$ for the truncated Szegő system [77], remains $C_u$ for the $\alpha$-deformation [75], and becomes $C_u - i\delta|(u|1)|^2(-2i\partial_\theta + \mathbf{I})$ for the $\delta$-deformation [77], where $\mathbf{I}$ is the identity operation. The classical conservation laws $\mathrm{Tr}[K_u^{2n}]$ are still present by the general Lax theory, while the presence or absence of extra conservation laws is understood less completely than for the original GG Hamiltonian. Overall, the presence of classical Lax pairs, especially where combined with an accurate Poissonian spectral statistics of the energy level spacings in the corresponding quantum theory, gives us confidence with using these systems as a pivotal reference point to represent generic quantum integrable systems in our studies.

# 6 Complexity bounds for quantum resonant systems

In section 4, we concluded that the upper bound on complexity (3.17) distinguishes integrable SYK models from chaotic ones very well. For the integrable SYK Hamiltonians, we observed that the lattice minimization methods described in section 3 managed to automatically locate lattice points separated from $E_n t$ in purely local directions, leading to length minimizer curves (in this case, exact geodesics) moving in local directions only. The lengths of these curves (and hence the corresponding complexity estimates) are then not impacted by the penalty factor $\mu$. This makes the complexity estimate grow more slowly with $D$ than what one expects in generic situations.

We stressed, however, that the strong reduction in the evolution complexity of integrable Hamiltonians compared to chaotic Hamiltonians found in section 4 relies crucially on the unique and peculiar structure of the integrable SYK models. The mild scaling of their complexity was pointed out in [21], where abundant purely local conservation laws with integer-valued spectra present in these models were used to construct very short paths from the unitary group identity to the evolution operator. Our lattice optimization techniques have succeeded in recovering and improving the results of these analytic constructions, while receiving no conscious human input about the analytic structure of the models. In generic integrable systems, however, it is not expected that large numbers of purely local conservation laws will exist (for a fixed small locality degree), even less so that their eigenvalue spectra will be exactly integer. We see no reasons to expect that the mechanisms of complexity reduction observed in the integrable SYK model will generalize broadly beyond the specific context of this model.

In this section, we therefore consider the upper bound on complexity for several chaotic and integrable Hamiltonians within the class of quantum resonant systems. There are a few advantages in using these models in our context. First, the integrable models we discussed in section 5.2 possess many features one expects from generic integrable systems. Although the original GG Hamiltonian is itself very non-generic with its purely integer energy spectrum, its deformations behave very much like textbook examples of quantum integrability with respect to their spectral statistics. Second, being bosonic, the models admit classical limits, where explicit Lax-pair structures are known for the integrable cases, with the corresponding towers of polynomial conservation laws. These towers are, however, not over-abundant at low polynomial degrees (which is what happens in SYK models), but have growing polynomial degrees much in the spirit of the classic integrability examples like the Korteweg-de Vries (KdV) equation, or one-dimensional nonlinear Schrödinger equation. Third, despite being bosonic, the models effectively operate in finite-dimensional Hilbert spaces, and furthermore the size of these Hilbert spaces grows much slower as a function of the particle number than it does in SYK models, providing more room for numerical experimentation.

We will treat the truncated Szegő model (5.11) to be the main representative for a 'generic' integrable system. In addition, we will also analyze the complexity of the $\alpha$- and $\delta$- deformations, (5.12) and (5.13), which inherit more or less 'superintegrable' structure from the GG Hamiltonian, depending on the values of the deformation parameters. As a representative chaotic system, we will be using random realizations of the quantum resonant Hamiltonian, as described in section 5.3.

## 6.1 Complexity analysis setup

The first step in deriving the complexity bound (2.30) is to determine the $Q$-matrix (2.26) that defines the distances to the lattice that parametrizes the sample curves (2.15-2.16). For this, we need to characterize the locality of operators acting on the Hilbert space of quantum resonant systems. Since the evolution operator is block-diagonal, we start by choosing a concrete $(N, M)$-block and restrict the discussion to states and operators within that block whose dimension is given by the number of integer partitions of $M$ into at most $N$ parts:

$$D = p_N(M),\tag{6.1}$$

as discussed in section 5.1. In practical applications we will focus on $N = M$, though it is essentially a matter of convenience.

Restricting to an $(N, M)$-block, one can build a basis out of operators of the form

$$a^\dagger_{n_1}\cdots a^\dagger_{n_N} a_{m_1}\cdots a_{m_N},\tag{6.2}$$

with $\sum n_i = \sum m_i = M$. Evidently, these operators map states from the $(N, M)$-block to the same block. One can easily verify as follows that these operators provide a full basis within that block. Since each Fock state $|a\rangle$ belonging to the $(N, M)$- block can be expressed as $A_J|0\rangle$, where $A_J$ is a monomial of degree $N$ in creation operators, the operators (6.2), normalized so as to satisfy (4.2), can be expressed as $|a\rangle\langle b|$. (Note that any $A^\dagger_J$ gives either 0 or something proportional to the vacuum $|0\rangle$ when acting on any state in the $(N, M)$-block, and hence $A_I|0\rangle\langle 0|A^\dagger_J = A_I A^\dagger_J$.) These operators clearly have a simple matrix representation in the Fock basis of the $(N, M)$-block. Since $|a\rangle\langle b|$ maps the Fock state $|b\rangle$ to the Fock state $|a\rangle$, the corresponding matrix has a single non-zero element. The number of choices of $|a\rangle$ and $|b\rangle$ is precisely the number of dimensions of the manifold of unitaries acting within the $(N, M)$-block. We remark that these operators are not Hermitian, although one can easily construct Hermitian combinations $|a\rangle\langle b| + |b\rangle\langle a|$ and $i(|a\rangle\langle b| - |b\rangle\langle a|)$. We will continue working with these non-Hermitian generators, which is more convenient and completely equivalent to using their Hermitian combinations.

A natural way to assign the locality degree of the operators (6.2) is by first removing all matching pairs of creation and annihilation operators (for the same mode) from the product expression and then counting the number of annihilation operators left, denoted $k$ below. This precisely indicates how many particles change their states under the action of the given operator. Under this convention, $k$-local operators precisely describe $k$-body interactions. Take an arbitrary quartic resonant Hamiltonian (5.1) as an illustration. This operator is 2-local since individual terms in the Hamiltonian involve interactions between 2 particles, moving these particles in new states. (There is a subset of terms in the resonant Hamiltonian that leaves the occupation numbers unchanged, and those are 0-local.)

The complete basis of operators $|a\rangle\langle b|$ can then be split into local and nonlocal entries according to a prescribed locality threshold $k$, above which an operator is considered hard or nonlocal. With these definitions for the generators $T_\alpha$ and $T_{\dot\alpha}$, the $Q$-matrix given by (2.26)

becomes[12]

$$Q_{nm} = \delta_{nm} - \sum_{(a,b) \in L} \langle n|a \rangle \langle b|n \rangle \langle m|b \rangle \langle a|m \rangle, \qquad (6.3)$$

where $L$ is the set of pairs of states $|a\rangle$ and $|b\rangle$ whose outer product $|a\rangle\langle b|$ is a local operator as defined by a given threshold $k$, and $|n\rangle$ are the energy eigenvectors as before.

Once this $Q$-matrix has been constructed, one simply proceeds with applying the lattice optimization techniques of section 3 to derive the complexity bound (3.17). This procedure will evidently produce different results for different locality thresholds $k$, and this dependence will be a key feature of our study, together with the dependence on the quantum Hamiltonian to which the bound is applied.

## 6.2 $Q$-matrices, Gram-Schmidt vectors and complexity profiles

In the complexity-related studies of [20, 21], the locality threshold was taken to be the locality degree of the Hamiltonian. In our present setup, where the Hamiltonian is a 2-body operator, this would amount to considering $k = 2$. While this approach is natural, there is no fundamental principle that mandates it, and we will be exploring a range of locality thresholds.

Integrable systems are typically characterized by a large number of conservation laws, and the general intuition, still far from having been made precise, is that this should somehow decrease the complexity values. In general, the conserved operators can be divided into local and nonlocal ones according to a specific chosen threshold. For the integrable quantum resonant models, restricting to local operators amounts to truncating the infinite tower of conservation laws to the first few entries, since the tower is made of polynomials of increasing degree in the creation-annihilation operators. At the same time, the saturation behavior of the quantum evolution complexity is determined by whether the vector $E_n t$ is typically close to a lattice point. This question is directly related to whether the $Q$-matrix possesses many low or zero eigenvalues, since this defines the directions in which the distance growth is slow. While the distinction between integrable and chaotic models may already be visible at locality $k = 2$, since the local conservation laws of integrable systems result in additional null vectors of the $Q$-matrix, the main differences between integrable and chaotic models are expected to appear in the movements of the $Q$-eigenvalues as one raises the locality threshold. This, in turn, will be reflected in the movements of the complexity plateau as one raises the locality threshold, and should lead to visible distinctions in the behavior of integrable and non-integrable systems. Our numerical analysis will largely validate this intuition.

We start by computing the eigenvalue distribution of the $Q$-matrix (6.3) for two integrable deformations of section 5.2 and one random instance of the general resonant Hamiltonian (5.1), whose coefficients are drawn from a uniform distribution over the interval $(0, 1)$, for different locality thresholds $k$. The results are displayed in Fig. 6. In contrast to the distributions found in Fig. 2 for the integrable SYK systems where the $Q$-eigenvalues were simply 0 and 1, the histograms for integrable quantum resonant models display a smeared distribution of eigenvalues. (We believe this situation to be generic, while the $Q$-spectrum of the integrable SYK models is a peculiarity.) A distinction between the integrable and chaotic distributions is nevertheless apparent. At $k = 2$, one observes for integrable systems a slightly larger spread around the main peak toward smaller values. This difference becomes much more pronounced as $k$ increases. The distributions of the integrable systems efficiently spread towards lower $Q$-eigenvalues (including an increasing number of exactly zero eigenvalues), which leads to an almost uniformly spread distribution at $k = 6$. The distributions of the chaotic models on

---

[12]Note that for non-Hermitian $T$'s, one should trade $T_\alpha$ for $T_\alpha^\dagger$ in one of the two factors in (2.26).

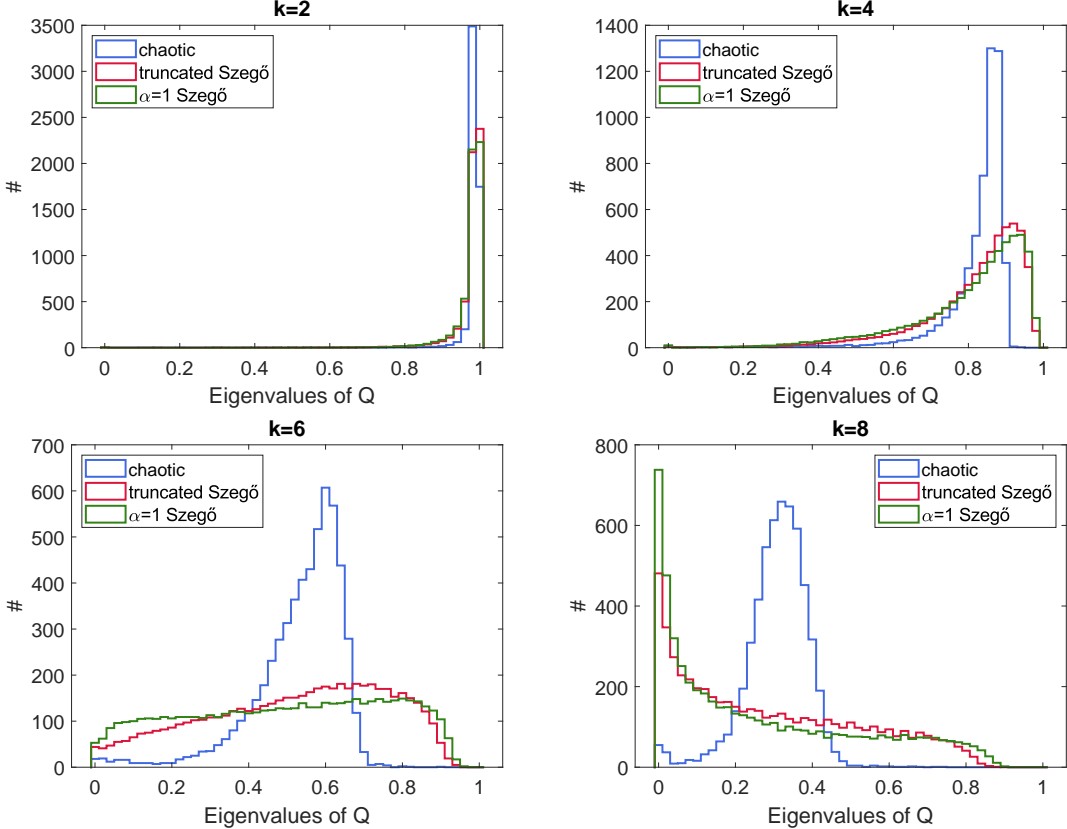

Figure 6: Histograms of the eigenvalues of the $Q$-matrix for a generic chaotic resonant Hamiltonian (blue), the integrable truncated Szegő model (5.11) (red) and the integrable $\alpha$-deformed Szegő model (5.12) (green) for different values of the locality threshold $k$ at $N = M = 30$.

the other hand remain peaked for all $k$, with predictably decreasing mean value. It is therefore important to notice that while the quantitative details of Fig. 2 and Fig. 6 are different, the qualitative observation that the $Q$-eigenvalues are generally lower for integrable models is present in both classes of models. This feature directly affects the complexity growth and saturation since it allows for a larger number of low-cost directions that can be explored in reaching out to nearby lattice points when minimizing (2.30).

Since our purpose is to compare the complexity for different systems and varying Hilbert space dimensions $D$, we impose the normalizations (2.19) and (2.20) on the Hamiltonian. Once the Hamiltonian and $Q$-matrix are determined for a quantum resonant system, one proceeds with the application of the basis reduction and lattice minimization techniques to approximately solve for the point of the lattice $2\pi\mathbb{Z}^D$ closest to $E_n t$ at every time step, which provides an upper bound on the complexity evolution following (3.17). First, we compute the LLL-reduced basis starting with the standard hypercubic basis in the metric $\mathbf{I} + (\mu - 1)\mathbf{Q}$, as reviewed in section 3.2. Then, we apply the nearest plane algorithm using the LLL-reduced basis as input and obtain an approximate solution to the CVP at a given time $t$. Finally, the resulting lattice point can be used as input for the greedy descent algorithm which may output an improved approximation for the lattice point closest to $E_n t$. In Fig. 7, we show the time evolution of the corresponding bound on complexity for different systems at $k = 2$ and $k = 6$. At early times, the complexity is predictably determined by the geodesic traced out by $e^{-itH}$ for all the systems, resulting in a unit slope linear growth. In some cases, this linear growth is interrupted by sudden short periods of complexity reduction, as can be observed for the lowest

two curves at $k = 6$. This behavior may originate from the existence of conserved quantities with an integer-valued spectrum, such as the operator (5.14). Depending on the details of the spectrum of the resonant Hamiltonian, these conservation laws can temporarily provide more advantageous directions in the manifold of unitaries. We note however that these observed intermittent decreases are not present for all the integrable systems and are therefore most probably not fundamental in nature.

The distinction between generic chaotic and integrable models is more evident in the saturation behavior of the complexity curves. The plateau height appears quite sensitive to the integrable structure of the model and a manifest hierarchy appears between chaotic and integrable curves. We find that the upper bound systematically assigns a larger complexity to chaotic models in accordance with the general expectation that chaotic evolution is inherently more complex than evolution constrained by integrability. Moreover, the hierarchy between chaotic and integrable systems becomes clearer as the locality threshold increases since heuristically one is exploiting more of the available integrable structure by allowing motion with no (or little) penalty in a larger subspace of operators.

Amongst the integrable models, the saturation height moreover seems to correlate with the degree of integrability of the model. Remember that in a given $(N, M)$-block, the $\delta$-deformed Szegő Hamiltonian is equivalent to an $\alpha$-deformation with $\alpha = M\delta$. When we described the level spacing statistics of quantum resonant models in Fig. 5, we pointed out that a small perturbation $\delta = 1/M$ (or $\alpha = 1$) retains noticeable features of the original GG Hamiltonian since it displays an excess of small energy spacings compared to Poissonian statistics. Similarly, when $\delta$ becomes too large, the trivial bilinear operator $a_0^\dagger a_0$ will start dominating the dynamics. In both limits, the Hamiltonian inherits a lot of structure from the nearby models that are more constrained than a generic integrable system, and one might expect the complexity to decrease further. At intermediate deformation values, the complexity of the $\delta$-deformation roughly coincides with the truncated Szegő model, which is our most generic representative for the integrable class.

These observations complement the complexity analysis applied to SYK models in section 4. For those models, the distinction between chaotic and integrable cases was extreme in that the functional dependence of the complexity on the Hilbert space dimension $D$ was entirely different. The slow scaling for the integrable models was identified in [21] and could be traced back to a peculiar profusion of local conservation laws. From the present analysis, we therefore conclude that the (upper bound on) complexity is not generally as strongly impacted for typical integrable models and this results in a moderate separation between chaotic and integrable models. Nevertheless, this separation is visibly sensitive to the imposed degree of locality $k$ and increases when $k$ increases. It remains an open question whether improving our bound and moving in the direction of the actual Nielsen's complexity will lead to more dramatic differences in plateau heights.



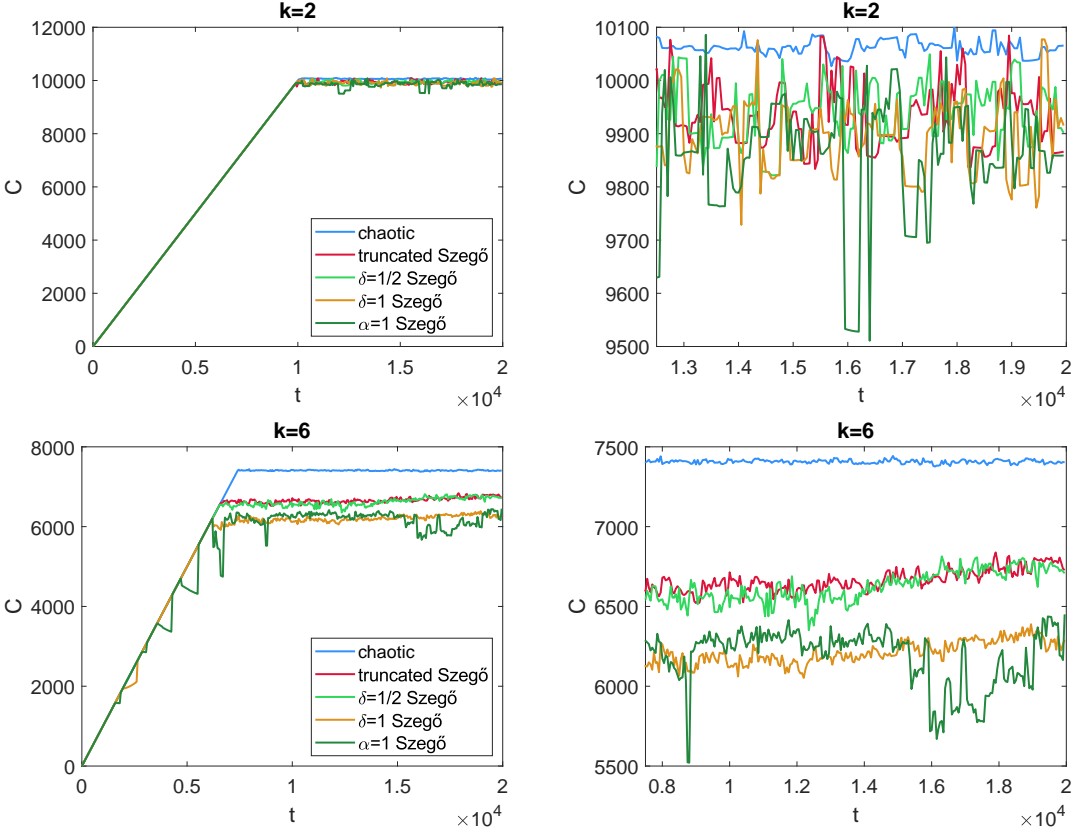

Figure 7: A comparison of the time evolution of the bound on complexity (3.17) at $N = M = 30$, where $D = 5604$, and locality threshold $k = 2$ for the top and $k = 6$ for the bottom figures, for five systems that display different degrees of integrability. **Left:** The complete time evolution of the complexity bound, where the early-time linear growth is visible, which corresponds to the situation where the shortest geodesic connecting the identity and $e^{-itH}$ is the physical trajectory itself. Although most systems follow this linear growth until saturation, some models, which typically possess a very constrained structure, seem to display periods of lowered complexity. **Right:** The same bounds on complexity are displayed at later times, focusing on the saturation behavior, which distinguishes systems according to their integrable features. From top to bottom the plateau values correspond to: a generic chaotic resonant Hamiltonian (blue), two generic integrable resonant Hamiltonians, i.e., the truncated Szegő model (5.11) (red) and $\delta = \frac{1}{2}$-deformed Szegő model (5.13) (lighter green), $\delta = 1$-deformed Szegő model (5.13) (yellow) and finally $\alpha = 1$-deformed Szegő model (5.12) (darker green) which does not display generic Poissonian statistics as can be seen in Fig. 5. We notice a clear separation in the plateau values between integrable and generic chaotic systems. It is also interesting to notice that the 'jitter' on the integrable plateau curves is more pronounced than for the generic chaotic model. This large variance becomes even more evident for the dark green curve and takes the form of recurring dips throughout the plateau region.

We proceed to systematically compute the scaling of the plateau heights of the complexity curves as a function of the Hilbert space dimension. To find the saturation values for the different models, we fix a time interval at late times and calculate the average complexity for times in $[50000, 54000]$ at snapshots separated by $\Delta t = 100$. (Since the complexity of the evolution associated to the $\alpha$-deformed Szegő model exhibited visible dips at these time scales,

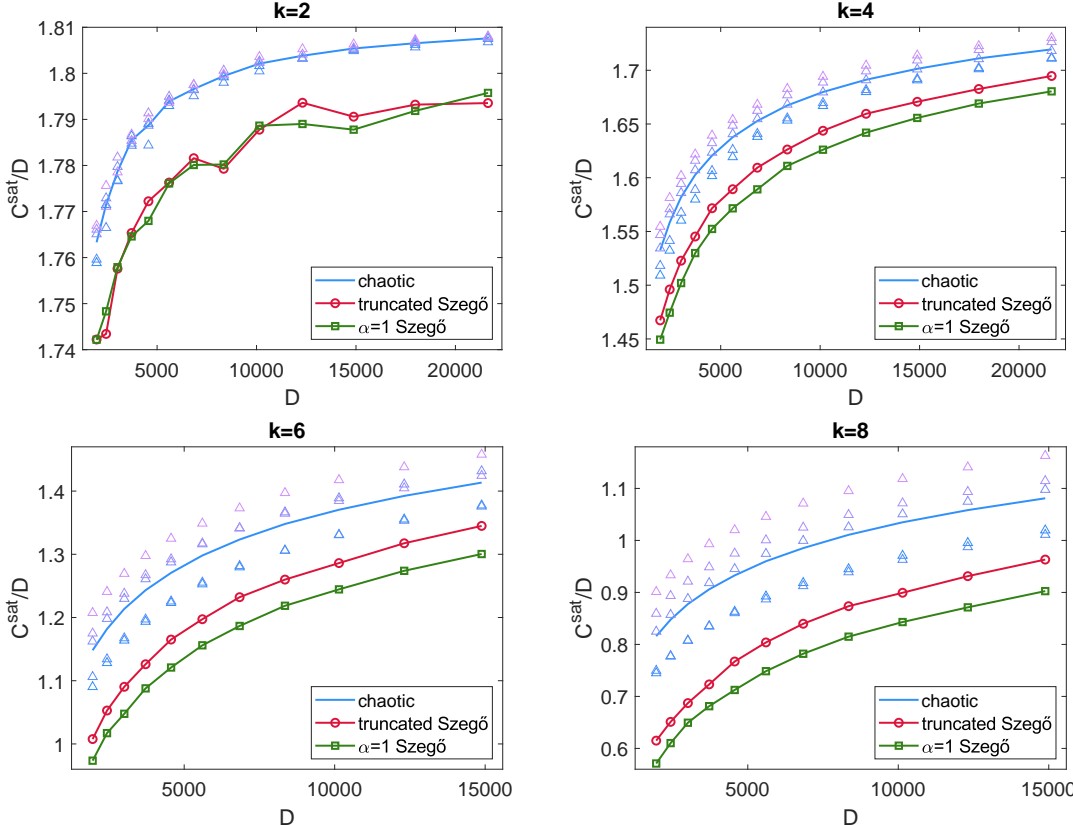

Figure 8: Saturation value of complexity divided by $D$ plotted against $D$ for the blocks $N = M = 25, \ldots, 37$ for $k = 2, 4$ and $N = M = 25, \ldots, 35$ for $k = 6, 8$. The blue line represents the average over five realizations of chaotic Hamiltonians which are denoted by triangles.

we used the interval $[5000000, 5004000]$ instead, where the complexity appeared to have saturated.) The resulting complexity plateau heights plotted against the Hilbert space dimension are displayed in Fig. 8. The simulations are performed for different locality thresholds $k$ within different $(N, N)$-blocks for five realizations of the chaotic Hamiltonian with random coefficients (kept fixed when moving from one Hamiltonian block to another) and two representative integrable models. At $k = 2$, which corresponds to the locality of the Hamiltonian, we notice a small spread in the complexity of the chaotic realizations and a small but clear separation between the integrable and chaotic systems. As $k$ increases, the spread in the saturation value amongst the chaotic models increases, and so does the separation between the integrable and chaotic models, maintaining a clearly visible hierarchy.

All the above figures display our best attempt at solving CVP involved in the complexity bound (2.30) at different moments in time by combining the LLL basis reduction algorithm, the nearest plane method and the greedy descent algorithm, in order to find an effective upper bound on quantum evolution complexity. Since no existing lattice minimization method is capable of solving CVP exactly in polynomial time, we put together a selection of techniques that work in synergy to provide a good approximation to the true solution. It is therefore interesting to ask about the relative contribution of the different methods by considering a few simplified algorithms to estimate the complexity bound. In Fig. 9, we compare the performance of several methods for a representative chaotic and integrable model. Leaving the more sophisticated methods momentarily aside, we first apply the simple rounding method described by (3.9), which finds the lattice point in $2\pi\mathbb{Z}^D$ closest to $E_n t$ in the standard Euclidean metric. While

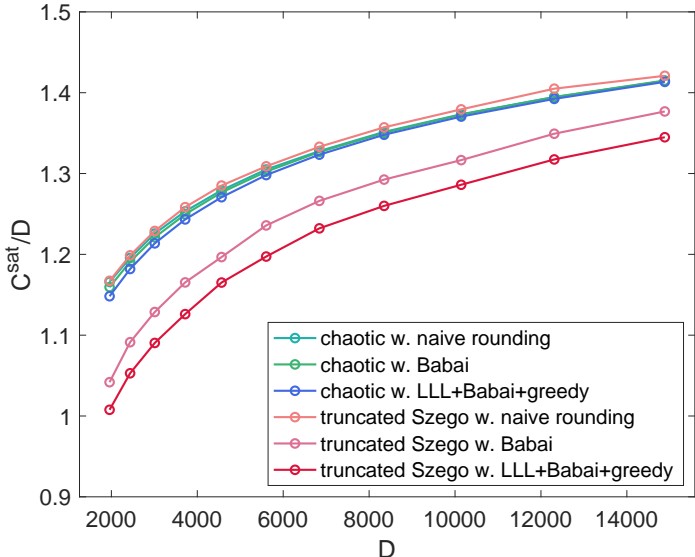

Figure 9: A comparison of different methods for solving the closest vector problem (CVP) that is central for the evaluation of the upper bound on complexity (2.30), as applied to a random resonant Hamiltonian and the truncated Szegő system, at $k = 6$.

this naive rounding method solves for the bi-invariant complexity exactly, Fig. 9 demonstrates that it does not perform well when $\mu \neq 1$. This rounding procedure is only sensitive to the properties of the energy eigenvalues and does not use any information about the eigenvectors, which are encoded in the metric $\mathbf{I} + (\mu - 1)\mathbf{Q}$. As a result, the associated complexity curves exhibit the worst performance of all and moreover fail at effectively distinguishing chaotic and integrable systems.

More refined methods, which do include information about the eigenvectors, succeed reasonably well reducing the complexity estimate for integrable vs. chaotic systems. Babai's nearest plane algorithm (without using an LLL-reduced basis) can be observed to significantly lower the complexity bound for integrable systems while the combination of the nearest plane algorithm using an LLL-reduced basis and the greedy descent algorithm improves the bound even further. In contrast, all possible combinations of these techniques only marginally improve the upper bound for the chaotic models, which may possibly mean that the bound is saturated and close to the true complexity value. These observations may have been expected from the distributions of the $Q$-eigenvalues showed in Fig. 6 since the distribution for generic chaotic models is consistently strongly concentrated about the mean, with the result that there are no real preferred directions on the lattice, in contrast to integrable models. It would be very interesting to identify the integrable properties that are at the origin of these differences and explore the consequences of this structure in more detail.

In section 3.4, we proposed an estimate for the late-time saturation value of the upper bound on complexity (2.30), which was obtained by generalizing to arbitrary lattices the interpretation of the plateau height of the bi-invariant complexity we discussed in section 2.2. Following the intuition that at late times the vector $E_n t$ represents a typical point in $\mathbb{R}^D$, the complexity would be expected to oscillate around the typical distance between a point in $\mathbb{R}^D$ and the nearest lattice point. If one then assumes that this average distance is concentrated around the maximal distance times $1/\sqrt{3}$, as for hypercubic lattices, an upper bound estimate (3.23) emerges in terms of the lengths of Gram-Schmidt vectors $||b_i^*||$. To verify the validity of this heuristic picture, we display the saturation values of the complexity bound for several systems and block sizes against the expected typical distance in the given lattice in Fig. 10

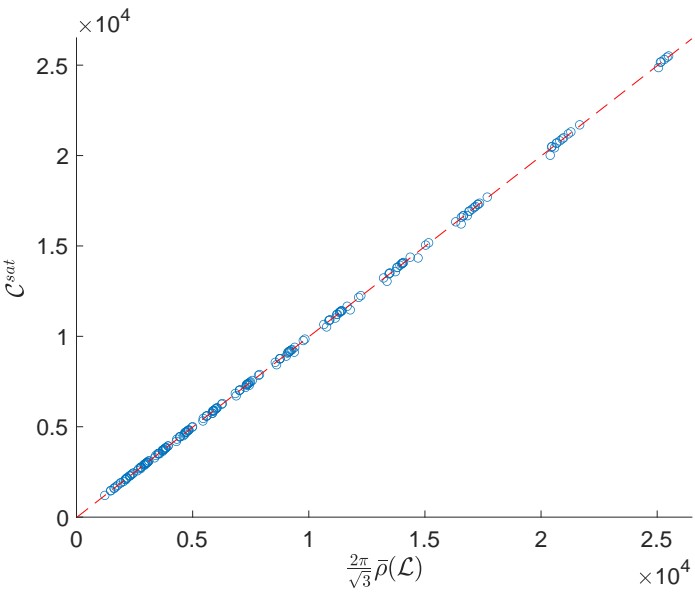

Figure 10: Numerically computed complexity saturation values for instances of the chaotic and truncated Szegő Hamiltonians with $N = M = 25, \ldots, 35$ and $k = 4, 6, 8$ plotted against the estimate (3.23). The red line is a linear fit to the data points with slope $\sim 0.9995$.

and find a very good correlation. Note that computing the upper bound (3.23) only involves elementary linear algebra.

Given the success of the estimate (3.23) in predicting the saturation value of our complexity measure, one could in principle avoid the trouble of going through the lattice minimization procedure at every time step and rely on the properties of the Gram-Schmidt vectors of the LLL-reduced basis to characterize chaotic and integrable models. This point of view gives some insight into the performance of the LLL algorithm. In Fig. 11 we show how the lengths of Gram-Schmidt orthogonalized basis vectors $||b_i^*||^2$ are modified by the application of the LLL agorithm. One can see that the algorithm achieves a noticeable reduction in the lengths of the Gram-Schmidt vectors for the integrable model, as one expects from the complexity reduction observed for integrable models and the efficiency of the estimate (3.23) with predicting the complexity plateaus, while it generally appears to be more effective for smaller Hilbert space dimensions. It is suggested by the considerations of [95] that the LLL algorithm works best for $D \lesssim 10^3$. Finding reduction algorithms that achieve a satisfactory trade-off between efficiency and run-time remains a challenge in the field of integer optimization. Nonetheless, for our purposes, the performance of the LLL algorithm is good enough since it results in efficient separation of generic (chaotic) models from systems with more analytic structure.

We finally make a preliminary observation on the oscillations around the complexity plateau. The amplitudes of these oscillations are visibly larger for integrable systems, which is a matter that clearly deserves further study. Similar increased oscillations around late-time plateaus for integrable systems have been observed for other quantities motivated by quantum chaos topics in [104].

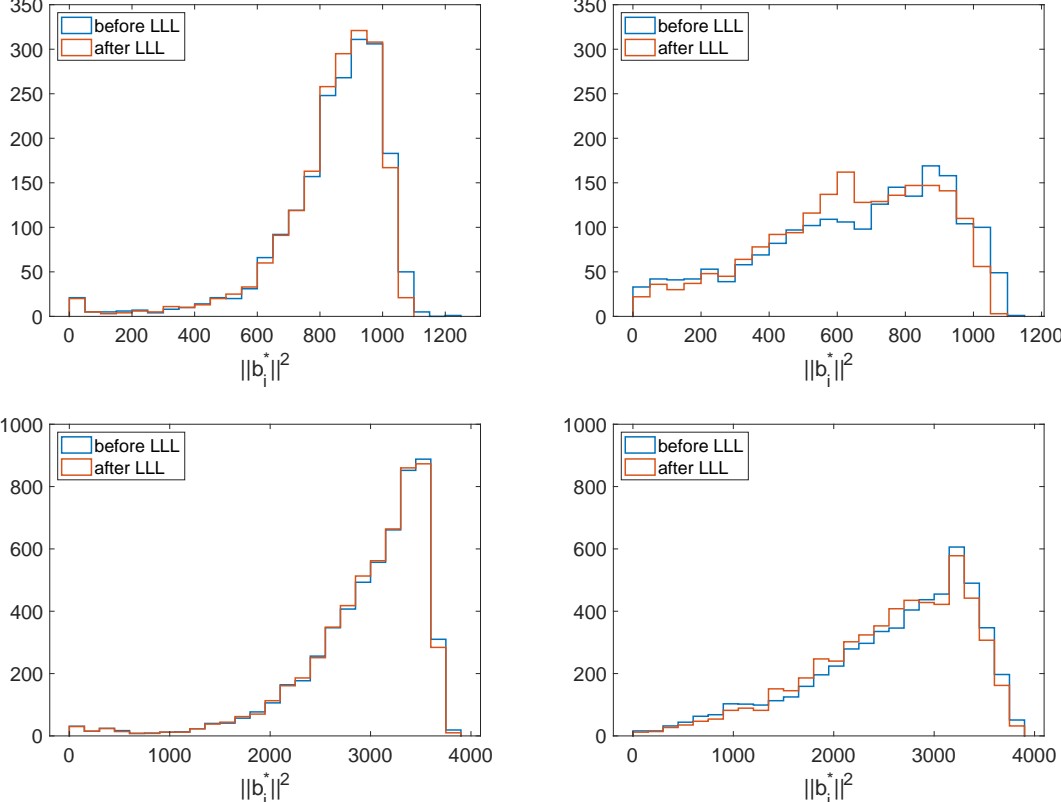

Figure 11: Histograms of the squared lengths of Gram-Schmidt orthogonalized basis vectors $b_i^*$ before and after the LLL reduction for **(top left)** a chaotic realization at $(N, M) = (25, 25)$ with $D = 1958$, **(top right)** truncated Szegő system at $(N, M) = (25, 25)$, **(bottom left)** a random quantum resonant Hamiltonian at $(N, M) = (30, 30)$ with $D = 5604$, **(bottom right)** truncated Szegő system at $(N, M) = (30, 30)$. The locality threshold was chosen as $k = 6$ in all four plots. One observes that the LLL basis reduction is more significant for the integrable model shown on the right than for the chaotic models on the left. This distinction directly impacts the corresponding complexity curves, since the estimate (3.23) for the plateau value is proportional to the average of the lengths of the Gram-Schmidt vectors, and leads to stronger complexity reduction for integrable models, as seen in Fig. 7 and Fig. 8.

# 7 Discussion

We have confronted the topic of Nielsen's complexity of quantum evolution for integrable and chaotic systems. While geometrically elegant and intuitive, the original definition of Nielsen's complexity is computationally intractable for Hilbert space sizes of physical interest. As an alternative, we proposed an upper bound on Nielsen's complexity given by (2.30). Our main question, which we answer in the affirmative, is whether this bound is by itself capable of distinguishing integrable and chaotic motion.

Evaluating the bound (2.30) is equivalent to the Closest Vector Problem (CVP), that is, given a point in Euclidean space to find the point of a given lattice closest to it. This problem has been widely studied in integer optimization theory, in particular, in relation to lattice-based cryptography. Exact solution of this problem is still intractable for Hilbert space sizes of physical interest, but there exists a range of established efficient polynomial-time algorithms that

find good suboptimal solutions. We have described how to practically use these algorithms to implement a numerical upper estimate on Nielsen's complexity. Using this implementation, we discover that for integrable fermionic SYK Hamiltonians it automatically recovers and improves the previously published bounds developed through dedicated system-specific analysis. We have furthermore applied the bound to quantum resonant systems, a family of Hamiltonians with some representatives that fit the profile of generic quantum integrability much closer than polynomial SYK models, and observed there as well that our bound systematically assigns lower values of complexity to integrable evolution operators.

We find that the plateau of the complexity upper bound scales with the Hilbert space dimension as $\mathcal{C}^{sat} \sim D$ for generic integrable and chaotic models, and distinguishes integrable from chaotic cases by displaying a smaller slope in the linear dependence on $D$ in the former case. (In the conventions used in [12, 20, 21], these results translate to rescaled saturation values $\mathcal{C}^{sat} \sim \sqrt{D}$.) The complexity reduction we see for integrable systems is rather modest (typically, say, at the level of 10-20%), though the separation of integrable and chaotic cases is clear. It remains an open question whether this is due to the difference of our upper bound from the true Nielsen's complexity, or it reflects the actual underlying situation. A similar modest reduction of complexity for integrable evolution has been recently observed in [105] for a different notion of quantum complexity. Fairly strong conjectures have been occasionally made in the literature in relation to the complexity reduction in integrable models, and observed for integrable SYK models in [21]. These latter models are, however, very special, as we explained at length in section 4, and one would not, *a priori*, expect these behaviors in more generic models.

In relation to the difference between the original Nielsen's complexity and our bound (2.30), the latter could be conceivably refined by attempting to run gradient descent in length starting from the curve (2.15-2.16) that acts as the actual minimizer in the complexity bound (2.30). Optimistically, one could hope that a good global optimization will be provided by our minimization over the infinite family of sample curves (2.15-2.16) winding in the manifold of unitaries in different ways, while the subsequent gradient descent will enhance it via local minimization. Running this procedure in practice would hinge on the ability to operate gradient descent effectively in very large numbers of dimensions.

It is a valid question how general one should consider our findings in section 6. While we have presented some solid arguments as to why integrable quantum resonant systems qualify as rather generic quantum integrable systems, admittedly our studies have been framed within the context of the concrete family of Hamiltonians (5.1). One important aspect is the presence of towers of polynomial conservation laws that provide exact null directions for the $Q$-matrix (2.26), so that there are directions within the optimization search in (2.30) in which the complexity growth is very slow. Polynomial conservation laws are a fairly generic feature of integrable systems, seen for example in the KdV equation or one-dimensional nonlinear Schrödinger equation. Some form of polynomiality is generally embedded into Lax integrability theory, since traces of powers of the first Lax operator provide conservation laws. Besides the exact zero eigenvalues, we observed in Fig. 6 strongly increased numbers of small eigenvalues for integrable cases, which likewise contributes to the complexity reduction. While it is hardly counterintuitive that the $Q$-eigenvalue distribution displays some level of continuity, and an increase in the number of zero eigenvalues comes together with an increase in the number of small eigenvalues, we do not have a direct explanation for this feature, and finding such explanation could be highly beneficial.

More broadly, the modest (but clear) reduction of complexity we have observed for the integrable models in section 6.2 can be traced back, qualitatively, to the eigenspectrum of the $Q$-matrix (2.26), since it determines how fast distances in the different directions of the minimization problem (2.30) grow. Through the estimate (3.23), which appears to hold rather

sharply in view of the empirical observations of Fig. 10, the properties of the complexity plateau are even more tightly related to the lengths of the Gram-Schmidt vectors corresponding to the lattice in the minimization problem (2.30). These Gram-Schmidt vectors are in turn completely determined by the $Q$-matrix. In this sense, any analytic understanding of $Q$-matrices originating from physical Hamiltonians, and the corresponding Gram-Schmidt vectors could considerably elucidate the functioning of our complexity bound (2.30) in application to physical evolution.

A more ambitious question is to understand how the behavior of complexity depends on the Hilbert space dimension and the locality threshold, especially for integrable systems. This is where one would hope to frame our empirical findings as sharp asymptotic statements one could attempt to prove. Such questions remain difficult at this moment, since one is limited numerically to Hilbert spaces of dimensions $10^4$ or perhaps slightly larger, but that only allows a few dozen particles in the context of quantum resonant systems (fewer in SYK models) and leaves little room for varying the locality threshold (which should physically be considerably below the number of particles).

Another avenue of exploration is spin chains, a very common arena for studies on quantum chaos topics. Here, even more freedom appears in defining complexity, as the notion of locality may now include spatial locality in addition to the many-body locality we have used in this paper. It will be intriguing to see how the sensitivity of complexity to physical information can be channelled and controlled by such choices.

# Acknowledgments

We thank Vijay Balasubramanian and Javier Magán for comments on an earlier version of the manuscript. This research has been supported by FWO-Vlaanderen projects G006918N and G012222N, and by Vrije Universiteit Brussel through the Strategic Research Program High-Energy Physics. MDC has been supported by a PhD fellowship from the Research Foundation Flanders (FWO). OE is supported by the CUniverse research promotion project (CUAASC) at Chulalongkorn University.

The computational resources and services used in this work were provided by the VSC (Flemish Supercomputer Center), funded by the Research Foundation Flanders (FWO) and the Flemish Government. In addition to the LLL algorithm implementation [94] we have used the partition generator [106].

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
