# Peer review of "Bounds on quantum evolution complexity via lattice cryptography"

_SciPost Physics, doi:SciPost Phys. 13, 090 (2022)_

## Round 1 · Referee Report · Anonymous (Referee 1) · 2022-5-31

Strengths

1- The paper is written in a self-contained fashion with appropriate exposition provided when relevant, e.g. on the Nielsen complexity, lattice optimization, and quantum resonant systems. 2- The level of detail provided on computations is excellent. It was exactly enough to essentially follow every line which made the paper's conclusions all the more convincing. This is a rare quality of paper writing in this subfield. 3- The paper demonstrates a clear awareness of prior art, has cited appropriately, has obviously built on prior results and makes precise distinctions as to what was done previously and what results are new. 4- The idea contained in the paper is very original and opens up a previously unexplored avenue for numerical studies of the behavior of the Nielsen complexity. 5- The actual writing quality is fantastic and a great model for scientific writing in general. Well done!

Weaknesses

1- Although there is a nice formulation of the upper bound for the complexity in terms of the Q-matrix I suppose it can be said that there is not as much analytic progress made on how the behavior of the Q-matrix relates to complexity as numerical. I for one would like to better analytically understand if the Q-matrix is really the relevant one, as opposed to the $R_{mn}$-matrix in [19, 20], and how its eigenvalue distribution relates to conservation laws. 2- As in previous work, there is an attempt to make conclusions about the behavior of generic integrable/chaotic systems based on a specific system/Hamiltonian. That being said, the authors make a genuine effort to justify their resonant systems as being more reflective of the generic integrable case.

Report

The paper is exceptionally clearly written and targets the important question as to whether there is a separation of time scales in the growth of the (Nielsen) complexity in quantum chaotic vs integrable systems.

The paper begins with a review of the formalism of the Nielsen complexity, in which the complexity (of time evolution, in this case) is defined as the length of the shortest geodesic from the identity to the time evolution operator on the unitary manifold in a certain metric that encodes the relative difficulty of different quantum operations. This section reformulates prior results on the bi-invariant metric (in which all operations have equal cost) by pointing out that the complexity is obtained by solving the closest vector problem on the Euclidean lattice. The principal observation that is the basis for much of the analysis in the remainder of this paper is that, when the metric is not bi-invariant, the geodesics of the bi-invariant metric provide an upper bound for the complexity when their length is evaluated in the new metric. This in turn amounts to solving the closest vector problem on a non-Euclidean lattice.

The bulk of the next part of the paper discusses several techniques that can be used to efficiently approximately solve the closest vector problem and iteratively improve these solutions. Complete implementation details are provided. These techniques do not provide exact solutions since the original problem is likely exponentially hard and indeed this hardness is the basis of lattice-based cryptography as alluded to by the title and discussion in the paper, so the numerical results of this paper are only approximate solutions to an upper bound for the complexity.

Nonetheless, the numerical results provided are nothing short of impressive. The first results consider a series of interacting Majorana fermion models that were first analyzed in refs. [19,20] in order to compare the numerical techniques to the results in those references, which essentially provided an exact solution for free models and a general upper bound for integrable models by analytically studying geometric properties of geodesics (conjugate points and geodesic loops). The numerical results in this paper closely follow the exact solution in the free case despite being produced by approximate algorithms for an upper bound, and display a remarkable reduction in the complexity in the integrable models. The authors study in detail how the eigenvalues of the "Q-matrix", which encodes the deviation of the metric from Euclidean, relate to local conservation laws, and show that these interacting Majorana fermion models are highly non-generic.

For this reason, the authors introduce quantum resonant systems and study the bounds on complexity in these models. These are block-diagonal quantum Hamiltonians that arise in weakly nonlinear interacting systems, which have well-defined classical limits with well-studied integrable dynamics in terms of conservation laws, Lax integrability, and more. The rough conclusion that one can draw from the results in this section of the paper is that generic integrable systems tend to have slightly decreased plateau heights (10%-20% as pointed out in the discussion), but in generic integrable systems these heights are not exponentially suppressed as indicated by previous work. The authors substantiate this surprising conclusion well by explicitly pointing out how the special structure of the Hamiltonians in previous work is responsible for the exponential suppression found previously, and examining carefully how the change in the eigenvalue distribution of the Q-matrix is responsible for the differing behaviors of the complexity (and in particular of the plateau height).

The authors of this paper have insightful analytic observations supplementing their numerical work, for instance by computing the expected plateau height for complexity curves by computing the average distance to a given lattice point, or by pointing out directions in the tangent space (4.25) that can generate short geodesic loops that lower the complexity in Majorana fermion models, which were missed in [19, 20].

This paper is an excellent addition to the quantum complexity literature that I highly recommend for publication. The observation that the Nielsen complexity computation can be presented as the closest vector problem in a non-Euclidean lattice is new and original and leads to new and unexpected conclusions as to the behavior of the complexity in different systems when studied numerically. There are potential avenues for future exploration pointed out in the discussion, for instance by studying the complexity in spin chains where incorporating the notion of spatial locality into the Nielsen metric may have direct consequences on the complexity. I repeatedly found myself writing comments for clarification or for future work only to find my comments/questions answered in the next paragraph or next page. A very thorough work!

Requested changes

1- There is a formatting error in LaTeX in the abstract which causes it to render improperly, please correct. 2- In Fig. 1, there is the statement that the variance of the red and blue curves on the right are comparable. This is very difficult to confirm visually and not an intuitive conclusion based on the plots of the curves, in which the blue curve seems to have much larger variance. It would be helpful here to add a plot of the distribution of deviations from the (time-averaged) mean for each curve, so that if it's the case that the blue curve is typically near the average but appears visually to have high variance due to occasional outliers, then that behavior is more apparent.

---

## Round 1 · Referee Report · Michal P. Heller (Referee 2) · 2022-8-2

Report

I read the article with great interest.

Study of complexity of time evolution in interacting quantum systems is a very difficult problem with only a few results available. The authors based their discussions on providing upper bounds on the behaviour of complexity by restricting to a clever choice of ansatzes for paths on the manifold of unitary operators. They observe the linear growth followed by a saturations, which are both the features predicted by rather high level arguments earlier in the literature.

The impressive aspect of the present studies are providing numerical results and some analytic understanding of the saturation value and the associated variance. In particular, the authors find an interesting transient difference between the variances for integrable and nonintegrable models within a one class and a small but measurable difference between the plateau values for the other class of models they considered.

The bottleneck of the present study is the need for diagonalization of the Hamiltonian in question, which limits the system size and makes it less likely that progress along these lines can be made on quantum field theory questions originating from holographic complexity proposals. However, this is a small price paid for the impressive results the authors are able to derive.

As far as I could see, the paper is correct and I am happy to recommend its acceptance.

---

## Editorial Decision

published